# SNPC-1.3 is a sex-specific transcription factor that drives male piRNA expression in *C. elegans*

Charlotte P Choi[1†], Rebecca J Tay[1†], Margaret R Starostik[1], Suhua Feng[2,3], James J Moresco[4], Brooke E Montgomery[5], Emily Xu[1], Maya A Hammonds[1], Michael C Schatz[1,6], Taiowa A Montgomery[5], John R Yates III[7], Steven E Jacobsen[2,8], John K Kim[1]*

[1]Department of Biology, Johns Hopkins University, Baltimore, United States; [2]Department of Molecular, Cell and Developmental Biology, University of California, Los Angeles, Los Angeles, United States; [3]Eli and Edythe Broad Center of Regenerative Medicine and Stem Cell Research, University of California, Los Angeles, Los Angeles, United States; [4]Center for the Genetics of Host Defense, University of Texas Southwestern Medical Center, Dallas, United States; [5]Department of Biology, Colorado State University, Fort Collins, United States; [6]Department of Computer Science, Johns Hopkins University, Baltimore, United States; [7]Department of Molecular Medicine, The Scripps Research Institute, La Jolla, United States; [8]Howard Hughes Medical Institute, University of California, Los Angeles, Los Angeles, United States

**\*For correspondence:**
jnkim@jhu.edu

†These authors contributed equally to this work

**Competing interests:** The authors declare that no competing interests exist.

**Abstract** Piwi-interacting RNAs (piRNAs) play essential roles in silencing repetitive elements to promote fertility in metazoans. Studies in worms, flies, and mammals reveal that piRNAs are expressed in a sex-specific manner. However, the mechanisms underlying this sex-specific regulation are unknown. Here we identify SNPC-1.3, a male germline-enriched variant of a conserved subunit of the small nuclear RNA-activating protein complex, as a male-specific piRNA transcription factor in *Caenorhabditis elegans*. SNPC-1.3 colocalizes with the core piRNA transcription factor, SNPC-4, in nuclear foci of the male germline. Binding of SNPC-1.3 at male piRNA loci drives spermatogenic piRNA transcription and requires SNPC-4. Loss of *snpc-1.3* leads to depletion of male piRNAs and defects in male-dependent fertility. Furthermore, TRA-1, a master regulator of sex determination, binds to the *snpc-1.3* promoter and represses its expression during oogenesis. Loss of TRA-1 targeting causes ectopic expression of *snpc-1.3* and male piRNAs during oogenesis. Thus, sexually dimorphic regulation of *snpc-1.3* expression coordinates male and female piRNA expression during germline development.

## Introduction

Piwi-interacting RNAs (piRNAs), a distinct class of small noncoding RNAs, function to preserve germline integrity (*Batista et al., 2008*; *Carmell et al., 2007*; *Cox et al., 1998*; *Deng and Lin, 2002*; *Kuramochi-Miyagawa et al., 2008*; *Lin and Spradling, 1997*; *Wang and Reinke, 2008*). In *Drosophila*, mutation of any of the three Piwi genes (*piwi, aub, ago3*) results in rampant activation of transposons in the germline and severe defects in fertility (*Brennecke et al., 2007*; *Harris and Macdonald, 2001*; *Lin and Spradling, 1997*; *Vagin et al., 2006*). In *Mus musculus*, mutation of the Piwi protein MIWI leads to the misregulation of genes involved in germ cell development, defective gametogenesis, and sterility (*Deng and Lin, 2002*; *Zhang et al., 2015b*). *Caenorhabditis elegans* piRNAs can

be inherited across multiple generations and trigger the transgenerational silencing of foreign elements such as transgenes. Disruption of this inheritance results in eventual germline collapse and sterility, known as the germline mortal phenotype (*Ashe et al., 2012*; *Buckley et al., 2012*; *Shirayama et al., 2012*). Taken together, piRNAs are essential to preserve germline integrity and ensure the reproductive capacity in metazoans.

Loss of the piRNA pathway can have distinct consequences between the sexes and across developmental stages. Many species show sex-specific expression of piRNAs (*Armisen et al., 2009*; *Billi et al., 2013*; *Williams et al., 2015*; *Yang et al., 2013*; *Zhou et al., 2010*). Demonstrated by hybrid dysgenesis, the identity of maternal, but not paternal, piRNAs in flies is important for fertility of progeny (*Brennecke et al., 2008*). In contrast, the piRNA pathway in mammals appears to be dispensable for female fertility (*Carmell et al., 2007*; *Murchison et al., 2007*), but distinct subsets of piRNAs are required for specific stages of spermatogenesis (*Aravin et al., 2003*; *Aravin et al., 2006*; *Carmell et al., 2007*; *Di Giacomo et al., 2013*; *Gainetdinov et al., 2018*; *Girard et al., 2006*; *Grivna et al., 2006*; *Kuramochi-Miyagawa et al., 2008*; *Li et al., 2013*). In worms, most piRNAs are uniquely enriched in either the male or female germline (*Billi et al., 2013*; *Kato et al., 2009*). Nevertheless, in all of these contexts, how the specific expression of different piRNA subclasses is achieved is poorly understood.

piRNA biogenesis is strikingly diverse across organisms and tissue types. In the *Drosophila* germline, piRNA clusters are found within pericentromeric or telomeric heterochromatin enriched for H3K9me3 histone modifications. The HP1 homolog Rhino binds to H3K9me3 within most of these piRNA clusters and recruits Moonshiner, a paralog of the basal transcription factor TFIIA, which, in turn, recruits RNA polymerase II (Pol II) to enable transcription within heterochromatin (*Andersen et al., 2017*; *Chen et al., 2016*; *Klattenhoff et al., 2009*; *Mohn et al., 2014*; *Pane et al., 2011*). Two waves of piRNA expression occur in mouse testes: pre-pachytene piRNAs are expressed in early spermatogenesis and silence transposons, whereas pachytene piRNAs are expressed in the later stages of meiosis and have unknown functions. While the mechanisms of pre-pachytene piRNA transcription remain elusive, pachytene piRNAs require the transcription factor A-MYB, along with RNA Pol II (*Li et al., 2013*).

In *C. elegans*, SNPC-4 is essential for the expression of piRNAs in the germline (*Kasper et al., 2014*). SNPC-4 is the single *C. elegans* ortholog of mammalian SNAPC4, the largest DNA binding subunit of the small nuclear RNA (snRNA) activating protein complex (SNAPc). A complex of SNAPC4, SNAPC1, and SNAPC3 binds to the proximal sequence element (PSE) of snRNA loci to promote their transcription (*Henry et al., 1995*; *Jawdekar and Henry, 2008*; *Ma and Hernandez, 2002*; *Su et al., 1997*; *Wong et al., 1998*; *Yoon et al., 1995*). SNPC-4 occupies transcription start sites of other classes of noncoding RNAs across various *C. elegans* tissue types and developmental stages (*Kasper et al., 2014*; *Weng et al., 2019*). Furthermore, piRNA biogenesis factors PRDE-1, TOFU-4, and TOFU-5 are expressed in germ cell nuclei and interact with SNPC-4 at clusters of piRNA loci (*Goh et al., 2014*; *Kasper et al., 2014*; *Weick et al., 2014*; *Weng et al., 2019*). These data suggest that SNPC-4 has been co-opted by germline-specific factors to transcribe piRNAs.

The vast majority of the ~15,000 piRNAs in *C. elegans* are encoded within two large megabase genomic clusters on chromosome IV (*Das et al., 2008*; *Ruby et al., 2006*). Each piRNA locus encodes a discrete transcriptional unit that is individually transcribed as a short precursor by Pol II (*Gu et al., 2012*; *Cecere et al., 2012*; *Billi et al., 2013*). Processing of precursors yields mature piRNAs that are typically 21 nucleotides (nt) in length and strongly enriched for a 5′ uracil (referred to as 21U-RNAs). Transcription of these piRNAs requires a conserved eight nt core motif (NNGTTTCA) within their promoters (*Billi et al., 2013*; *Cecere et al., 2012*; *Ruby et al., 2006*). piRNAs enriched during spermatogenesis are associated with a cytosine at the 5′ most position of the core motif (CNGTTTCA); mutation of cytosine to adenine at this position results in ectopic expression of normally male-enriched piRNAs during oogenesis. In contrast, genomic loci expressing piRNAs enriched in the female germline show no discernable nucleotide bias at the 5′ position (*Billi et al., 2013*). While differences in cis-regulatory sequences contribute to the sexually dimorphic nature of piRNA expression, sex-specific piRNA transcription factors that drive distinct subsets of piRNAs in the male and female germlines remain to be identified.

Here, we demonstrate that SNPC-1.3, an ortholog of human SNAPC1, is required specifically for male piRNA expression. Furthermore, TRA-1, a master regulator of sex determination, transcriptionally represses *snpc-1.3* during oogenesis to restrict its expression to the male germline. Taken

together, our study reports the first example of a sex-specific piRNA transcription factor that drives the expression of male-specific piRNAs.

## Results

### SNPC-4 is a component of the core piRNA transcription complex that drives all piRNA expression

SNPC-4-specific foci are present in both male and female germ cell nuclei (*Kasper et al., 2014*), but the role of SNPC-4 in the male germline is not well understood. We hypothesized that SNPC-4 is required for piRNA biogenesis in both the male and female germline. To test this, we conditionally depleted the SNPC-4 protein using the auxin-inducible degradation system (*Zhang et al., 2015a*; *Figure 1—figure supplement 1A*). We added an auxin-inducible degron (AID) to the C-terminus of SNPC-4 using CRISPR/Cas9 genome engineering, and crossed this strain into worms expressing TIR1 under the germline promoter, *sun-1*. TIR1 is a plant-specific F-box protein that mediates the rapid degradation of *C. elegans* proteins tagged with an AID in the presence of the phytohormone auxin. Thus, addition of auxin to the *snpc-4::aid*; *Psun-1::TIR1* strain is expected to degrade SNPC-4::AID, whereas strains with *snpc-4::aid* alone serve as a negative control; under these conditions, we examined a panel of spermatogenesis- and oogenesis-enriched piRNAs (*Billi et al., 2013*) during spermatogenesis and oogenesis. Unless otherwise stated, spermatogenesis and oogenesis stages will correspond to time points taken at 48 hr and 72 hr, respectively, post-L1 hatching at 20°C. Worms depleted of SNPC-4 showed decreased expression of both spermatogenesis- and oogenesis-enriched piRNAs during spermatogenesis and oogenesis time points, respectively (*Figure 1A*), confirming that SNPC-4 is a core piRNA transcription factor required for all piRNA expression.

Given that SNPC-4 activates transcription of piRNAs in both sexes, we hypothesized that sex-specific cofactors might associate with SNPC-4 to regulate sexually dimorphic piRNA expression. To test this hypothesis, we leveraged genetic backgrounds that masculinize or feminize the germline. Specifically, we used *him-8(-)* mutants, which have a higher incidence of males (~30% males compared to <0.5% spontaneous males in the wild-type hermaphrodite population) (*Hodgkin et al., 1979*), and *fem-1(-)* mutants, which are completely feminized when grown at 25°C (*Doniach and Hodgkin, 1984*). We introduced a C-terminal 3xFlag tag sequence at the endogenous *snpc-4* locus using CRISPR/Cas9 genome editing (*Paix et al., 2015*) and performed immunoprecipitation of SNPC-4::3xFlag followed by mass spectrometry. PRDE-1 and TOFU-5 co-purified with SNPC-4::3xFlag in both *him-8(-)* and *fem-1(-)* mutants, suggesting that these known piRNA biogenesis factors exist as a complex in both male and female germlines (*Figure 1B,C, Figure 1—figure supplement 1B*). While a single worm ortholog, SNPC-4, exists for human SNAPC4, the *C. elegans* genome encodes four homologs of human SNAPC1 (worm SNPC-1.1, -1.2, -1.3, and -1.5) and four homologs of human SNAPC3 (worm SNPC-3.1, -3.2, -3.3, and -3.4, *Figure 1B*; *Li et al., 2004*). From our mass spectrometry analysis, six of the eight *C. elegans* homologs of SNAPC1 and SNAPC3 co-purified with SNPC-4::3xFlag from both *him-8(-)* and *fem-1(-)* genetic backgrounds (*Figure 1B,C*). These results revealed that SNPC-4 interacts with both snRNA and piRNA transcriptional machinery.

### SNPC-1.3 interacts with the core piRNA biogenesis factor SNPC-4 during spermatogenesis

We also identified proteins that co-purified with SNPC-4::3xFlag from *him-8(-)*, but not *fem-1(-)* mutants. We were particularly interested in SNPC-1.3 because of its homology to the mammalian SNAPC1 subunit of the snRNA transcription complex. We confirmed that SNPC-1.3 interacts with SNPC-4 by using CRISPR/Cas9 genome editing to generate an endogenously tagged *snpc-1.3::ollas* strain. We then crossed *snpc-1.3::ollas* into the *snpc-4::3xflag* strain and performed immunoprecipitation with anti-Flag antibodies. In agreement with the mass spectrometry data, SNPC-4::3xFlag and SNPC-1.3:Ollas interacted robustly during spermatogenesis. The interaction was detectable at a much lower level during oogenesis (*Figure 1D*). The reciprocal co-immunoprecipitation of SNPC-1.3::3xFlag followed by western blotting for SNPC-4::Ollas confirmed this biochemical interaction (*Figure 1—figure supplement 1H*), suggesting that SNPC-1.3 forms a complex with the previously characterized piRNA biogenesis factor SNPC-4.

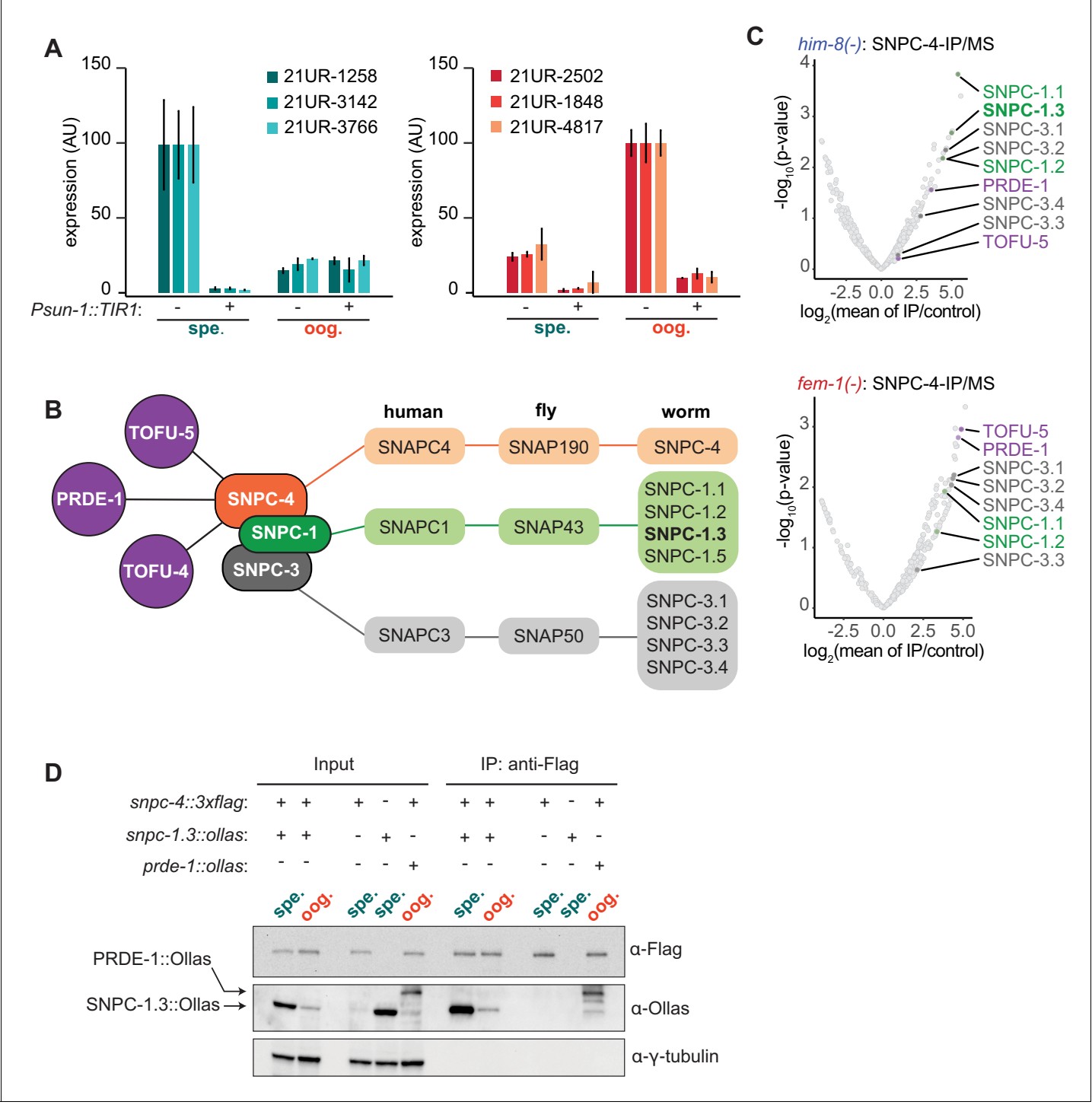

**Figure 1.** SNPC-4 and SNPC-1.3 are part of the male piRNA transcription complex. (A) SNPC-4 is required for both male and female piRNA expression. Taqman qPCR of male (left) and female (right) piRNAs normalized to U18 small nucleolar RNA in *snpc-4::aid* (denoted as '−') and *snpc-4::aid; Psun-1::TIR1* (denoted as '+') worms. Both genotypes were placed on auxin, and collected during spermatogenesis (spe., 48 hr) and oogenesis (oog., 72 hr). Error bars: ± SD from two technical replicates. (B) Schematic highlights the conservation of SNAPc homologs from *C. elegans*, *D. melanogaster, and H. sapiens* and catalogs all SNPC-4 (orange) interacting partners from previous work (*Weick et al., 2014*; *Weng et al., 2019*) or from our own analysis. Known piRNA biogenesis factors (purple), SNPC-1 paralogs (green), and SNPC-3 paralogs (gray) are indicated. (C) SNPC-1.3 interacts with SNPC-4 in only *him-8(-)* mutants. Volcano plots showing enrichment values of IP of SNPC-4 over control (control: *him-8(-)* mutants for top panel or *fem-1(-)* mutants for bottom panel) and analogous significance values for proteins that co-purified with SNPC-4::3xFlag from (top) *him-8(-)* mutants or (bottom) *fem-1(-)* mutants (n = 2 biological replicates). piRNA biogenesis factors (purple), SNPC-1 paralogs (green), and SNPC-3 paralogs (dark gray) are labeled in (B).
*Figure 1 continued on next page*

*Figure 1 continued*

Although SNPC-3.1 and SNPC-3.2 are reported to have the same amino acid sequence, we have picked up differential peptide coverage in the *fem-1(-)* mutant for these two proteins and represented them as two different data points. (**D**) SNPC-4 interacts with SNPC-1.3. Anti-Flag immunoprecipitation of SNPC-4::3xFlag and western blot for SNPC-1.3::Ollas during spermatogenesis (spe.) and oogenesis (oog.). PRDE-1::Ollas was used as a positive control for interaction with SNPC-4::3xFlag (*Kasper et al., 2014*). γ-Tubulin was used as the loading control.

The online version of this article includes the following source data and figure supplement(s) for figure 1:

**Source data 1.** Source data for *Figure 1A*.

**Figure supplement 1.** Validation of strains and mass spectrometry.

## SNPC-1.3 is enriched in the male germline

To determine whether SNPC-1.3 expression is restricted to the germline, we first examined *snpc-1.3* mRNA levels during early spermatogenesis (36 hr post-L1 hatching) in worms fed *glp-1* RNAi, which abrogates germline development. The mRNA of *snpc-4*, which is highly expressed in the germline (*Kasper et al., 2014*), was used as a control. Knockdown of *glp-1* mRNA markedly reduced both *snpc-1.3* and *snpc-4* mRNAs. SNPC-1.3::3xFlag protein expression was also reduced in a *glp-4* temperature-sensitive mutant, which fails to develop fully expanded germlines at 25°C (*Beanan and Strome, 1992*), suggesting that SNPC-1.3 is predominantly expressed in the germline (*Figure 2A*).

To examine differential *snpc-1.3* expression between the sexes, we measured *snpc-1.3* mRNA levels in *him-8(-)* males and *fem-1(-)* females. The expression of *snpc-1.3* mRNA was greatly enriched in *him-8(-)* relative to *fem-1(-)*, while *snpc-4* mRNA did not show any differential expression. At the protein level, SNPC-1.3::3xFlag was also highly enriched in males as compared to females by western blotting (*Figure 2B*).

SNPC-4, along with other piRNA factors, such as PRDE-1, localize to one or two foci in each germline nuclei (*Kasper et al., 2014*; *Weick et al., 2014*; *Weng et al., 2019*). Given that SNPC-1.3 is present in a complex with SNPC-4 (*Figure 1D*), we hypothesized that SNPC-1.3 might show a similar localization pattern to these other piRNA factors. To examine the subcellular localization of SNPC-1.3, we performed immunofluorescence in *snpc-4::3xflag; snpc-1.3::ollas* adult males and hermaphrodites. In the male germline, SNPC-1.3::Ollas colocalized with SNPC-4::3xFlag in the same nuclear foci (*Figure 2C*). In contrast, no SNPC-1.3::Ollas signal was detected above background in hermaphrodites (*Figure 2C*). Taken together, these data indicate that SNPC-1.3 co-localizes with SNPC-4 specifically in the male germline.

## SNPC-1.3 is required for transcription of male piRNAs

Given the prominent interaction between SNPC-1.3 and SNPC-4 in the male germline (*Figure 1D*), we hypothesized that SNPC-1.3 might be required for piRNA expression during spermatogenesis. To test this hypothesis, we generated a *snpc-1.3* null allele by introducing mutations that result in a premature stop codon located eight amino acids away from the start codon at the *snpc-1.3* locus. We examined spermatogenesis in hermaphrodites and *him-8(-)* males and examined oogenesis in adult hermaphrodites and *fem-1(-)* females. As a control, we analyzed the loss-of-function mutant of the *C. elegans* Piwi protein, *prg-1(-)*, which almost completely lacked male and female piRNAs (*Figure 3A*), as expected. Levels of male piRNAs were dramatically reduced in *snpc-1.3(-)* hermaphrodites during spermatogenesis and in *him-8(-); snpc-1.3(-)* males, whereas female piRNAs were largely unaltered in *snpc-1.3(-)* adult hermaphrodites and in *fem-1(-); snpc-1.3(-)* females (*Figure 3A,B*). Unexpectedly, female piRNAs were also moderately upregulated by at least twofold in *snpc-1.3(-)* mutants undergoing spermatogenesis and in *him-8(-); snpc-1.3(-)* males. These findings suggest that, in addition to activating male piRNAs, SNPC-1.3 suppresses the expression of female piRNAs in the male germline, possibly by preferentially recruiting core factors such as SNPC-4 to male piRNA loci. As SNPC-4 is known to activate transcription of snRNAs as well as piRNAs (*Kasper et al., 2014*), we asked whether SNPC-1.3 is also required for transcribing snRNAs. To test this, we measured U1 snRNA levels in hermaphrodite adults after RNAi-mediated knockdown of *snpc-1.3*. In contrast to the reduction of U1 observed in *snpc-4* RNAi, U1 levels were not significantly altered when *snpc-1.3* was depleted (*Figure 3—figure supplement 2A*), suggesting that, unlike SNPC-4, SNPC-1.3 is likely specific to the transcription of male piRNAs and does not play a role in snRNA transcription.

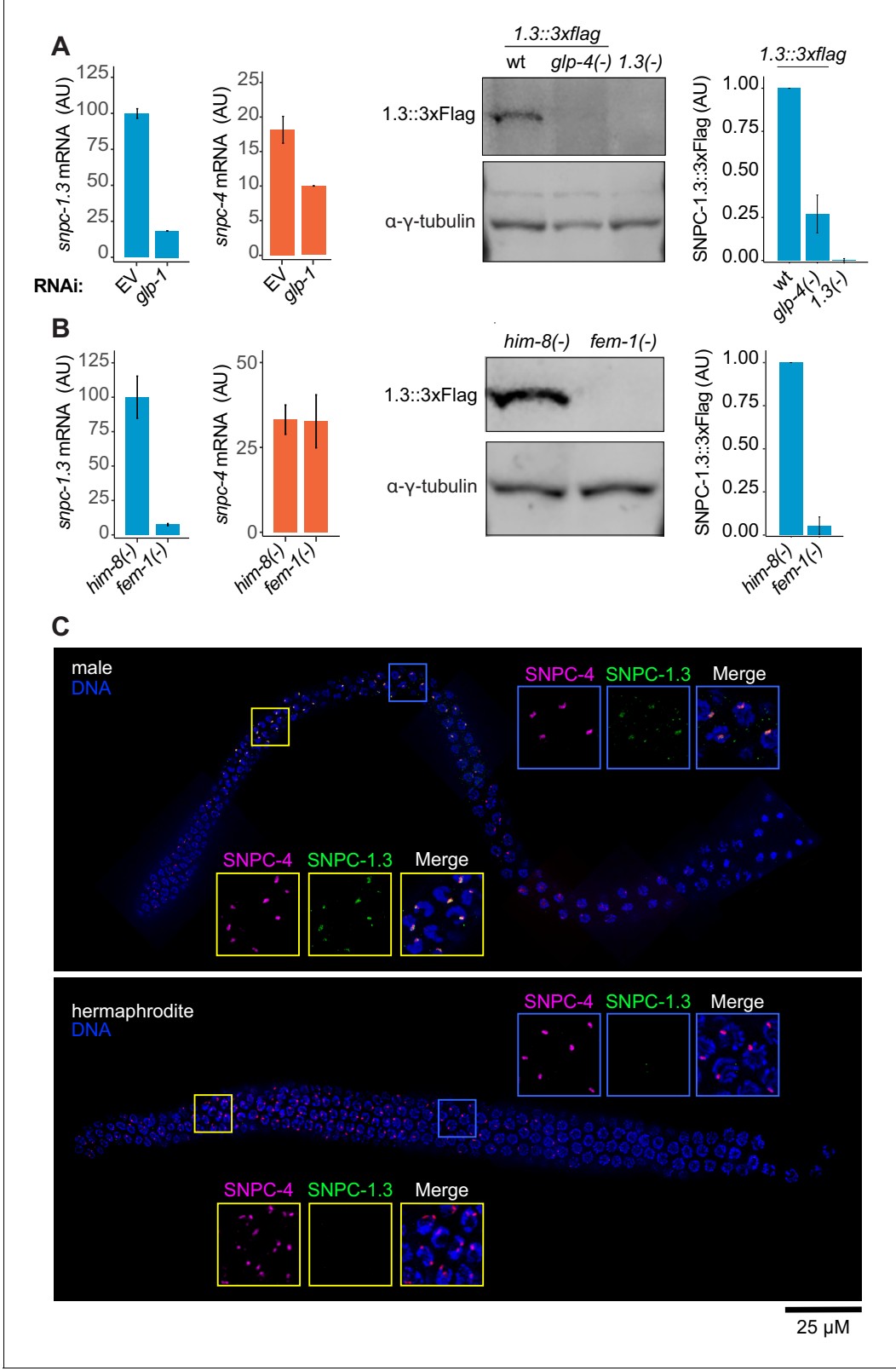

**Figure 2.** SNPC-1.3 is enriched in the male germline. (**A**) SNPC-1.3 is predominantly germline-expressed. (Left) *snpc-1.3* mRNA expression is reduced upon RNAi-mediated knockdown of *glp-1* during early spermatogenesis (36 hr). The housekeeping gene *eft-2* was used for normalization. Error bars: ± SD of two technical replicates. (Right) Western blot and quantification of SNPC-1.3::3xFlag in wild type, *glp-4(-)*, and *snpc-1.3(-)* (no-Flag control) during spermatogenesis. Error bars: ± SD of two biological replicates. γ-Tubulin was used as the loading control. (**B**) SNPC-1.3 is more highly expressed in
*Figure 2 continued on next page*

*Figure 2 continued*

males. (Left) *snpc-1.3* mRNA expression is dramatically enriched in *him-8(-)* males over *fem-1(-)* females during spermatogenesis, whereas *snpc-4* mRNA expression shows no specific enrichment. *eft-2* was used for normalization. Error bars: ± SD of two technical replicates. (Right) Western blot and quantification of SNPC-1.3::3xFlag in *him-8(-)* and *fem-1(-)*. Error bars: ± SD of two biological replicates. γ-Tubulin was used as the loading control. (C) SNPC-1.3 colocalizes with SNPC-4 in the male germline. Dissected adult male (top) and hermaphrodite (bottom) germlines stained for DNA, SNPC-4::3xFlag (magenta) and SNPC-1.3::Ollas (green) in a N2 background. Yellow insets: transition zone. Blue insets: pachytene. Representative image of three biological replicates is shown (male, n = 21, 18, 15 and hermaphrodite, n = 18, 10, 10). Scale bar, 25 μm.

The online version of this article includes the following source data for figure 2:

**Source data 1.** Source data for *Figure 2A*.
**Source data 2.** Source data for *Figure 2A*.
**Source data 3.** Source data for *Figure 2B*.
**Source data 4.** Source data for *Figure 2B*.

To extend these findings, we identified piRNAs enriched during spermatogenesis and oogenesis by small RNA-seq in wild-type worms. Using a 1.2-fold threshold and false discovery rate (FDR) of ≤0.05, a total of 6,368 of 14,714 piRNAs on chromosome IV were differentially expressed (*Figure 3C*; *Figure 3—figure supplements 1* and *2C*; *Supplementary file 1*). Among these, 4,060 piRNAs were upregulated during spermatogenesis (hereafter referred to as male piRNAs) and 2,308 piRNAs were upregulated during oogenesis, which we define as female piRNAs. We compared this dataset with our previous study that identified and categorized spermatogenesis- and oogenesis-enriched piRNAs, as well as piRNAs that were not statistically enriched (NE) either during oogenesis or spermatogenesis (*Billi et al., 2013*). Most male piRNAs identified in this study were also identified in our previous study (82%; 3,316/4,060; *Figure 3C*). Next, we investigated how loss of *snpc-1.3* affects global piRNA expression by performing small RNA-seq in wild type versus *snpc-1.3(-)* mutants during spermatogenesis. We identified 3,601 piRNAs that were downregulated in a *snpc-1.3(-)* mutant compared to wild type (*Figure 3D*, *Figure 3—figure supplement 2D*, *Supplementary file 2*). Of these, 3,002 overlapped with spermatogenesis-enriched piRNAs identified in our previous study (*Billi et al., 2013*; *Figure 3D*). Additionally, 85% (3,452/4,060) of male piRNAs were depleted in *snpc-1.3(-)* mutants, suggesting that male piRNAs are regulated by SNPC-1.3 (*Figure 3E*, *Figure 3—figure supplement 2E*). Consistent with our Taqman analysis (*Figure 3A,B*), 73% (1,687/2,308) of oogenesis-enriched piRNAs identified in our study were significantly upregulated in *snpc-1.3(-)* mutants during spermatogenesis (*Figure 3F*).

We next analyzed the genomic loci of male piRNAs and *snpc-1.3*-dependent piRNAs. As expected, the intersection of these two piRNA subsets displayed strong enrichment for the eight nt core motif and the 5'-most position of this core motif was enriched for cytosine (CNGTTTCA. *Figure 3E*; *Figure 3—figure supplement 2E*). In contrast, the core motif found upstream of female piRNAs upregulated upon loss of *snpc-1.3* displayed a much weaker bias for the 5' cytosine (*Figure 3F*). These observations validate our previous findings that male and female core motifs are distinct (*Billi et al., 2013*). Taken together, these data indicate that SNPC-1.3 is required for male piRNA expression.

## SNPC-1.3 binds male piRNA loci in a SNPC-4-dependent manner

Given that SNPC-1.3 interacts with SNPC-4 and is required for expression of male piRNAs, we hypothesized that SNPC-1.3 might bind male piRNA loci in association with SNPC-4. To test this, we performed ChIP-qPCR to investigate SNPC-1.3 occupancy at regions of high piRNA density within the two large piRNA clusters on chromosome IV; an intergenic region lacking piRNAs served as a control. To determine whether SNPC-1.3 binding was dependent on SNPC-4, we again used the auxin-inducible degradation system to deplete SNPC-4 in the *snpc-1.3::3xflag* strain for 4 hr prior to our spermatogenesis time point. In the presence of SNPC-4 expression, SNPC-1.3 was enriched at both piRNA clusters, albeit to a lesser degree at the small cluster, and this enrichment was lost upon SNPC-4 depletion (*Figure 4A*, *Figure 4—figure supplement 1A*). These data indicate that SNPC-1.3 binds piRNA loci during spermatogenesis in a SNPC-4-dependent manner in vivo.

To examine the genome-wide binding profile of SNPC-1.3 and its dependency on SNPC-4, we performed ChIP-seq of N2, *snpc-1.3::3xflag*, and *snpc-1.3::3xflag; snpc-4::aid; Psun-1::TIR1* worms

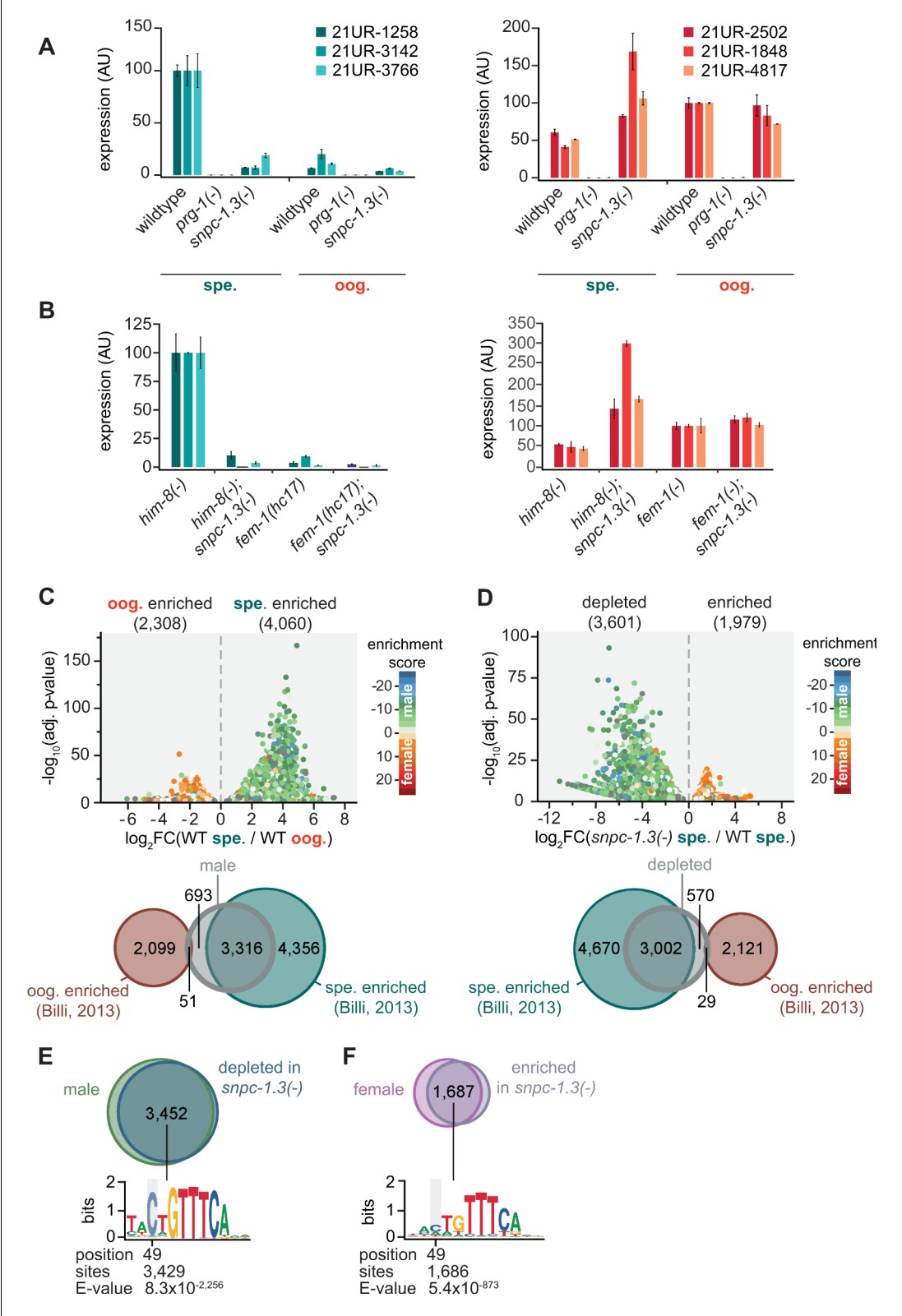

**Figure 3.** SNPC-1.3 is required for transcription of male piRNAs. (**A**) *snpc-1.3* is required for male piRNA expression (spe.) but is dispensable for female piRNA expression during oogenesis (oog.). Taqman qPCR and quantification of representative male (left) and female (right) piRNAs at spermatogenic and oogenic time points normalized to U18. Error bars: ± SD of two technical replicates. (**B**) *him-8(-); snpc-1.3(-)* mutant males exhibit severely impaired male piRNA expression and enhanced female piRNA expression. *snpc-1.3* is not required for male or female piRNA expression in *fem-1(-)* females.
*Figure 3 continued on next page*

*Figure 3 continued*

Error bars: ± SD from two technical replicates. (C) piRNAs are differentially expressed during spermatogenesis (spe.) and oogenesis (oog.) in wild-type worms. (Top) Volcano plot showing piRNAs with ≥1.2 fold-change and FDR of ≤0.05 in 48 hr (spe.) versus 72 hr (oog.). piRNAs are colored according to male and female enrichment scores from *Billi et al., 2013*. (Bottom) Overlap of male piRNAs (spe.) in wild type at 48 hr with spermatogenesis-enriched and oogenesis-enriched piRNAs defined in *Billi et al., 2013*. (D) piRNAs depleted in *snpc-1.3(-)* comprise mostly of male piRNAs. (Top) Volcano plot shows piRNAs with ≥1.2 fold-change and FDR ≤ 0.05 in *snpc-1.3(-)* mutant versus wild type during spermatogenesis (spe.). piRNAs are colored according to male and female enrichment scores from *Billi et al., 2013*. (Bottom) Overlap of *snpc-1.3*-dependent piRNAs with spermatogenesis- and oogenesis-enriched piRNAs defined in *Billi et al., 2013*. (E) Male piRNAs that are depleted in *snpc-1.3(-)* have a conserved upstream motif with a strong 5' C bias. (Top) Overlap of *snpc-1.3*-dependent piRNAs with male piRNAs shown in (C). (Bottom) Logo plot displays conserved motif upstream of each piRNA. Median position of the C-nucleotide of the identified motif, number of piRNAs, and associated E-value are listed. (F) Female piRNAs are upregulated in *snpc-1.3(-)* mutants during spermatogenesis. (Top) Overlap of piRNAs upregulated at 72 hr (oog.) with piRNAs enriched in *snpc-1.3(-)* at 48 hr (spe.). (Bottom) Logo plot displays conserved motif upstream of each piRNA. Median position of the C-nucleotide of the identified motif, number of piRNAs, and associated E-value are listed.

The online version of this article includes the following source data and figure supplement(s) for figure 3:

**Source data 1.** Source data for *Figure 3A*.
**Source data 2.** Source data for *Figure 3B*.
**Figure supplement 1.** Small RNA-seq analysis pipeline.
**Figure supplement 2.** Quality control of small RNA-seq and validation analysis.

during spermatogenesis (*Figure 4—figure supplement 1B,C*). Consistent with our ChIP-qPCR results, SNPC-1.3 binds piRNA clusters in a SNPC-4-dependent manner (*Figure 4B*, *Figure 4—figure supplement 1D*). By quantifying the SNPC-1.3 signal over consecutive, non-overlapping 1 kb bins across the entire genome, we identified 691 1 kb regions within the chromosome IV piRNA clusters that were enriched for SNPC-1.3 in *snpc-1.3::3xflag* compared to N2 (*Figure 4C*, *Figure 4—figure supplement 1G*). Relative to *snpc-1.3::3xflag*, worms depleted of SNPC-4 showed loss of SNPC-1.3 in 749 1 kb regions on chromosome IV piRNA clusters (*Figure 4D*, *Figure 4—figure supplement 1F*). Furthermore, SNPC-1.3 enrichment ($p<2.2\times10^{-16}$) and depletion ($p<2.2\times10^{-16}$) were specific to the piRNA clusters on chromosome IV, and more than half (393/691) of the SNPC-1.3-enriched regions in *snpc-1.3::3xflag* worms were depleted upon degradation of SNPC-4 (*Figure 4C,D*, *Figure 4—figure supplement 1F,G*).

To determine whether SNPC-1.3 preferentially binds male piRNA loci, we characterized the SNPC-1.3 signal around individual 5' nucleotides of mature piRNAs. Again, we classified piRNAs as male, female, or not significantly enriched (NE) in either sex, based on our small RNA-seq analysis in wild-type hermaphrodites during spermatogenesis and oogenesis (*Figure 3C*). SNPC-1.3 binding at male piRNA loci was most enriched just upstream of the piRNA 5' nucleotide, which overlaps the conserved core motif (*Figure 4E*, *Figure 4—figure supplement 1E*). This binding profile was very distinct for 1 kb bins that contained only male piRNAs (*Figure 4F*, *Figure 4—figure supplement 1H*). Upon depletion of SNPC-4, this peak in male piRNAs was lost (*Figure 4E*, *Figure 4—figure supplement 1E*). Although the binding profiles for individual female piRNAs exhibited more variability, there was little evidence for SNPC-1.3 binding and dependency on SNPC-4 at female loci (*Figure 4E*, *Figure 4—figure supplement 1E*). Compared to the binding profile in male piRNA loci, SNPC-1.3 binding was observed to a lesser extent in non-enriched piRNAs (*Figure 4E*, *Figure 4—figure supplement 1E*). Taken together, these observations indicate that SNPC-1.3 requires the core piRNA factor SNPC-4 to bind the piRNA clusters during spermatogenesis.

## TRA-1 represses *snpc-1.3* and male piRNA expression during oogenesis

As male piRNA expression and SNPC-1.3 protein expression are largely restricted to the male germline, we asked how *snpc-1.3* mRNA expression is regulated across development. *C. elegans* hermaphrodites produce sperm during the L4 stage and transition to producing oocytes as adults. To understand the mRNA expression profile of *snpc-1.3* relative to *snpc-4* and other developmentally regulated genes, we performed qRT-PCR across hermaphrodite development. *snpc-4* mRNA is expressed at low levels during spermatogenesis, but dramatically increases during oogenesis (*Figures 5A* and *1D*; *Figure 1—figure supplement 1H*). These data suggest that low levels of SNPC-4 are sufficient for activating male piRNA biogenesis during spermatogenesis. Consistent with SNPC-1.3 protein expression (*Figure 1D*), *snpc-1.3* mRNA levels peak in L3 to early L4 stages,

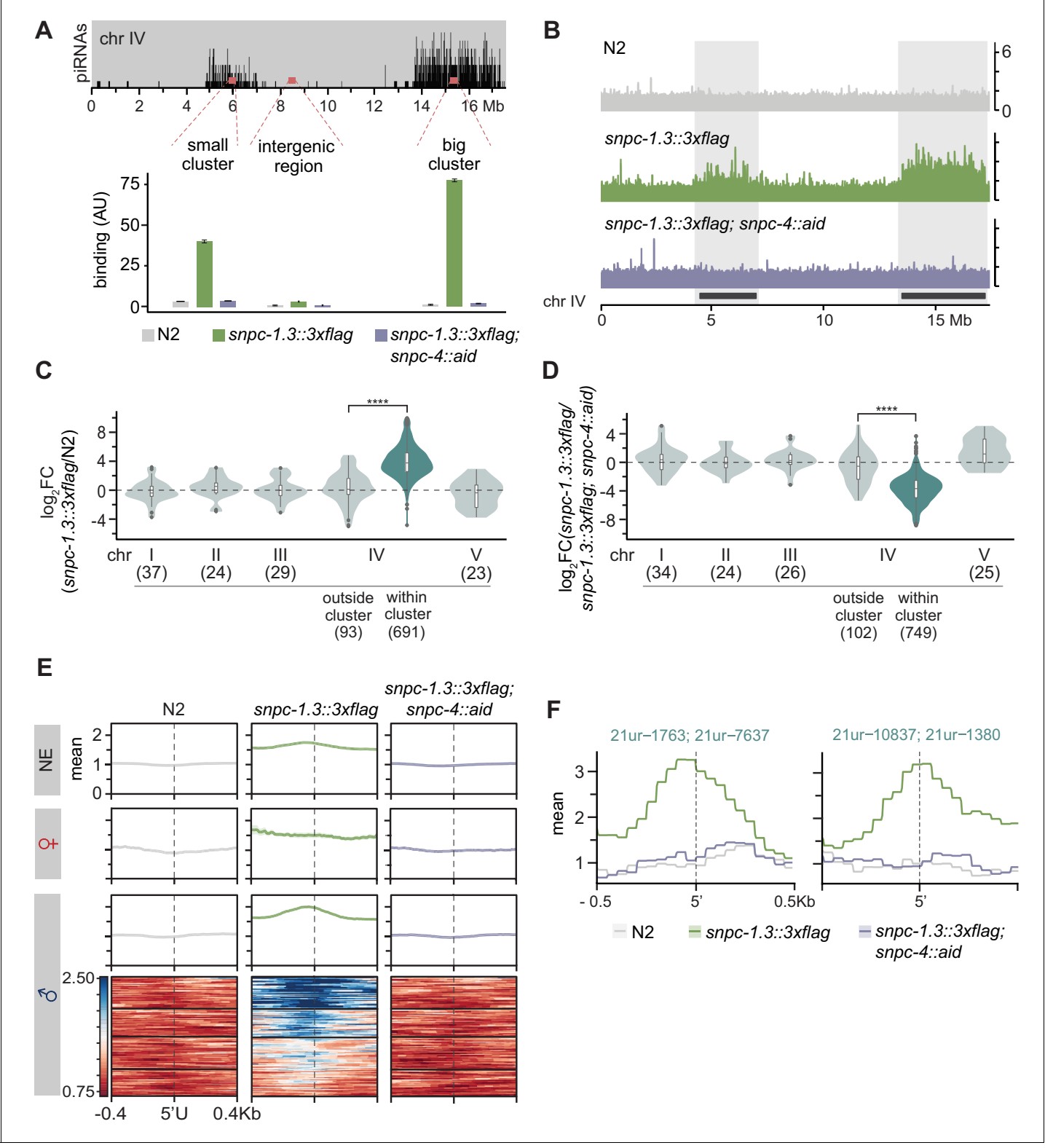

**Figure 4.** SNPC-1.3 binds male piRNA loci in a SNPC-4-dependent manner. (**A**) SNPC-1.3 binding at the piRNA clusters requires SNPC-4. SNPC-1.3::3xFlag binding normalized to input (mean ± SD of two technical replicates) on chromosome IV by ChIP-qPCR in N2, *snpc-1.3::3xflag*, and *snpc-1.3::3xflag; snpc-4::aid::ollas*, which undergoes TIR-1-mediated degradation by addition of auxin (*snpc-4::aid*). Top panel depicts the density of piRNAs on chromosome IV with piRNAs predominantly found in the small (4.5–7 Mb) and big (13.5–17.2 Mb) clusters. (**B**) SNPC-1.3 binding profiles across chromosome IV in N2, *snpc-1.3::3xflag*, and *snpc-1.3::3xflag; snpc-4::aid*. The locations of the two piRNAs clusters are highlighted. (**C**) SNPC-1.3 binding

*Figure 4 continued on next page*

*Figure 4 continued*

is enriched at piRNA clusters on chromosome IV. SNPC-1.3-bound regions are enriched within piRNA clusters compared to regions outside of the piRNA clusters on chromosome IV (****p≤0.0001, Wilcoxon rank sum test). The number of bins analyzed is listed in parentheses. (D) SNPC-1.3 enrichment at piRNA clusters is dependent on SNPC-4. SNPC-1.3-bound regions within piRNA clusters are depleted compared to regions outside of the piRNA clusters on chromosome IV upon loss of SNPC-4 (****p≤0.0001, Wilcoxon rank sum test). The number of bins analyzed is listed in parentheses. (E) Distribution of SNPC-1.3 reads (mean density ± standard error) around the 5′ nucleotide of mature piRNAs at the piRNA clusters. To resolve SNPC-1.3 binding between male and female piRNAs despite the high density of piRNAs, we selected 1 kb bins with all male (100), female (19), or non-enriched (279) piRNAs. Heat maps represent ChIP signal in 1 kb bins around the 5′ nucleotide of all 100 mature male piRNAs, ranked according to SNPC-1.3 signal. (F) Examples of SNPC-1.3 binding at two regions containing two male piRNA loci. Regions are anchored on the 5′ nucleotide of each mature male piRNA and show mean read density ± standard error.

The online version of this article includes the following source data and figure supplement(s) for figure 4:

**Source data 1.** Source data for *Figure 4A*.
**Figure supplement 1.** SNPC-1.3 ChIP-seq pipeline and quality control and biological replicates for SNPC-1.3 ChIP.

during spermatogenesis (*Figure 5A*). Given that *snpc-1.3* expression across development is regulated at the mRNA level, we examined the sequences upstream of the *snpc-1.3* coding region to identify potential *cis*-regulatory motifs. Less than 200 bp upstream of the *snpc-1.3* start codon, we identified three consensus binding sites for TRA-1 (*Figure 5B*), a transcription factor that controls the transition from spermatogenesis to oogenesis (*Berkseth et al., 2013*; *Clarke and Berg, 1998*; *Zarkower and Hodgkin, 1993*).

In the germline, TRA-1, a Gli family zinc-finger transcription factor, controls the sperm-to-oocyte decision by repressing both *fog-1* and *fog-3*, which are required for controlling sexual cell fate (*Berkseth et al., 2013*; *Chen and Ellis, 2000*; *Lamont and Kimble, 2007*; *Zarkower and Hodgkin, 1993*). Loss-of-function *tra-1* hermaphrodites exhibit masculinization of the female germline and develop phenotypically male-like traits (*Hodgkin, 1987*). We used RNAi to knock down *tra-1* and observed significant ectopic upregulation of *snpc-1.3* mRNA during oogenesis (*Figure 5C*). However, this upregulation of *snpc-1.3* expression could be an indirect effect of masculinization of the germline. Therefore, to test whether TRA-1 directly regulates *snpc-1.3*, we generated strains harboring mutations at the three TRA-1 binding sites (*tbs*) in the endogenous *snpc-1.3* promoter. Specifically, we mutated one (*1xtbs*), two (*2xtbs*), or all three (*3xtbs*) consensus TRA-1 binding motifs (*Figure 5B*). Disruption of the TRA-1 binding sites led to reduced TRA-1::3xFlag binding upstream of *snpc-1.3* as revealed by ChIP-seq, with the *3xtbs* mutant showing the greatest reduction of binding (*Figure 5B*, *Figure 5—figure supplement 1D*). In addition, *snpc-1.3* mRNA levels were highly upregulated when multiple TRA-1 binding sites were mutagenized (*Figure 5C*), consistent with TRA-1 directly repressing *snpc-1.3* transcription during oogenesis. To confirm that SNPC-1.3 protein expression was also elevated in TRA-1 binding site mutants, we used CRISPR/Cas9 engineering to add a C-terminal 3xFlag tag at the *snpc-1.3* locus in *snpc-1.3 (2xtbs)* mutants. Indeed, SNPC-1.3::3xFlag showed increased expression in the *snpc-1.3::3xFlag(2xtbs)* mutant during spermatogenesis and especially oogenesis (*Figure 5C*). Taken together, these findings demonstrate TRA-1 binds to the *snpc-1.3* promoter to repress its transcription during oogenesis.

Given that *snpc-1.3* is robustly de-repressed during oogenesis in TRA-1 binding site mutants, we hypothesized that male piRNAs would also be ectopically upregulated during oogenesis. To test this, we performed small RNA-seq and compared piRNA levels in wild-type and *snpc-1.3 (2xtbs)* worms during oogenesis (*Supplementary file 3*). Using a 1.2-fold threshold and FDR of ≤0.05, we observed 1,370 piRNAs in *snpc-1.3 (2xtbs)* mutants that were upregulated compared to wild type (*Figure 5D*). The majority of these upregulated piRNAs overlap with the male piRNAs that we identified in wild-type hermaphrodites (*Figure 5D*). We also confirmed this result by Taqman qPCR analysis, which showed that male piRNAs were significantly upregulated in *snpc-1.3 (2xtbs)* and *snpc-1.3 (3xtbs)* mutants compared to wild type during oogenesis (*Figure 5E*). Taken together, these data suggest that TRA-1 directly binds to *tbs* sites in the *snpc-1.3* promoter to repress its transcription and consequently, male piRNA expression during oogenesis.

Our data showed that female piRNAs are inappropriately upregulated during spermatogenesis upon loss of *snpc-1.3* (*Figure 3A*). Consistent with this result, female piRNAs show reduced expression during oogenesis upon upregulation of SNPC-1.3 expression in *snpc-1.3 (2xtbs)* and *snpc-1.3 (3xtbs)* mutants compared to wild type (*Figure 5E*). We posit that SNPC-1.3 plays a direct role in

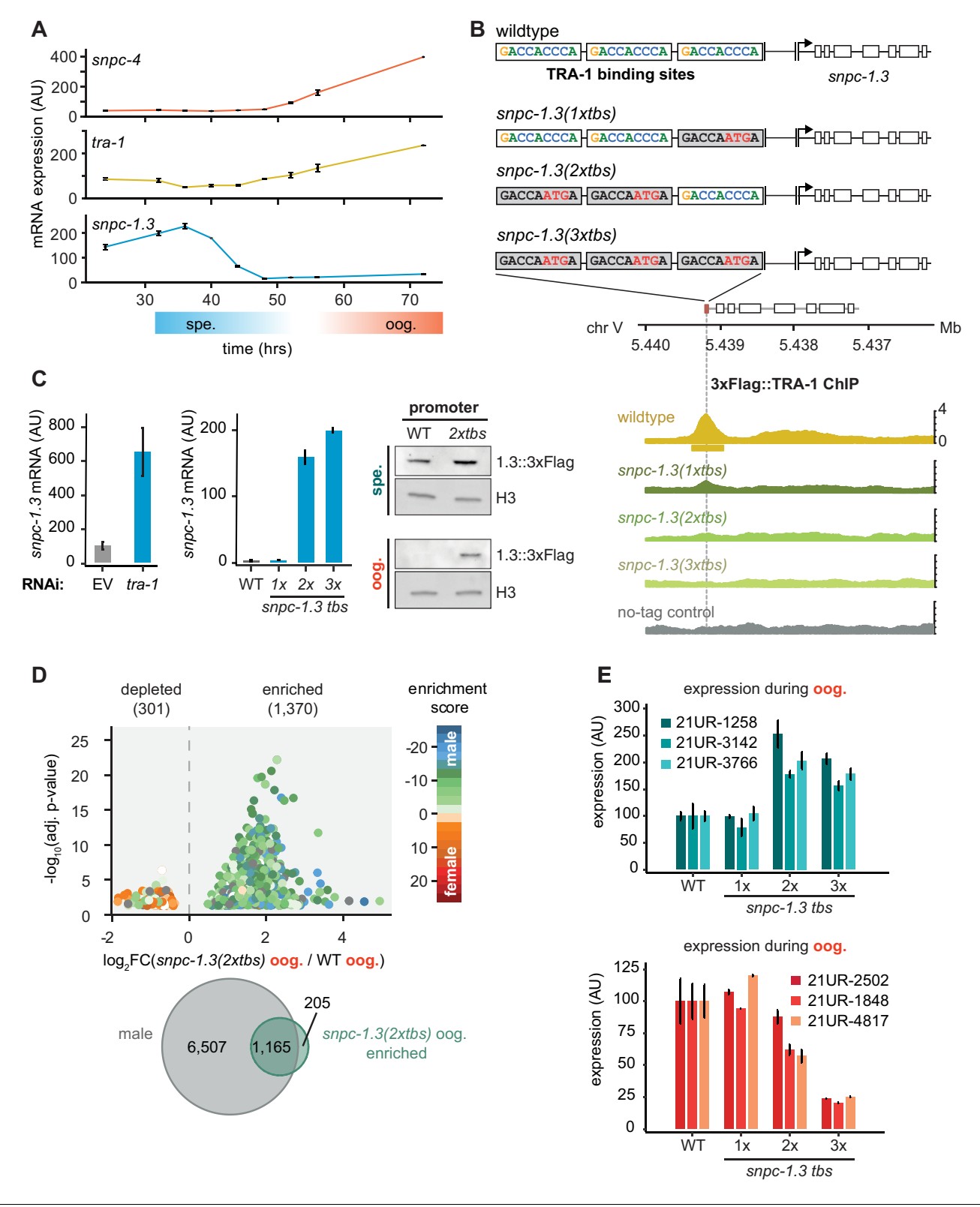

**Figure 5.** TRA-1 represses *snpc-1.3* and male piRNAs expression during oogenesis. (**A**) *snpc-1.3* mRNA levels peak during early spermatogenesis (spe.) while *tra-1* mRNA levels are highest during oogenesis (oog.). qRT-PCR and quantification of *snpc-1.3*, *snpc-4*, and *tra-1* mRNA normalized to *eft-2* mRNA across hermaphrodite development. Time zero corresponds to the time when synchronized L1s were plated. Error bars: ± SD of two technical replicates. (**B**) TRA-1 binds to the *snpc-1.3* promoter. Schematic of the three TRA-1 binding sites upstream of the *snpc-1.3* locus in wild type (top). Site-
*Figure 5 continued on next page*

Figure 5 continued

specific mutations shown in red were made in one, two, or three of the TRA-1 binding sites (gray denotes the mutated motifs). (Bottom) TRA-1 binding is reduced in TRA-1 binding site mutants assayed by TRA-1 ChIP-seq. (C) TRA-1 represses *snpc-1.3* mRNA expression during oogenesis. (Left) *snpc-1.3* mRNA expression is drastically upregulated upon RNAi-mediated knockdown of *tra-1* and (middle) in strains bearing mutations in two (*2xtbs*) or three (*3xtbs*) TRA-1 binding sites. Error bars indicate ± SD from two technical replicates. (Right) Western blot of SNCP-1.3::3xFlag expression driven under the wild-type and *2xtbs* mutant promoter during spermatogenesis (spe.) (top) and oogenesis (oog.) (bottom). H3 was used as the loading control. (D) A subset of male piRNAs are ectopically expressed during oogenesis in *snpc-1.3 (2xtbs)* mutants. (Top) Volcano plot showing differential piRNA expression between *snpc-1.3 (2xtbs)* mutants versus wild type during oogenesis (oog.). piRNAs are colored by enrichment scores from *Billi et al., 2013*. (Bottom) Overlap of male piRNAs defined in *Figure 3C* with upregulated piRNAs in *snpc-1.3 (2xtbs)* mutants. (E) Mutations at two (*2xtbs*) or three (*3xtbs*) TRA-1 binding sites enhance male piRNA expression (top) but attenuate female piRNA expression (bottom) during oogenesis. Error bars indicate ± SD from two technical replicates.

The online version of this article includes the following source data and figure supplement(s) for figure 5:

**Source data 1.** Source data for *Figure 5A*.
**Source data 2.** Source data for *Figure 5C,E*.
**Figure supplement 1.** TRA-1 regulation of *snpc-1.3* across hermaphrodite development.

activating male piRNA transcription, while indirectly limiting female piRNA transcription by sequestering core piRNA transcription factors to male piRNA loci.

## SNPC-1.3 is critical for male fertility

Given the global depletion of male piRNAs in *snpc-1.3(-)* mutants and the progressive fertility defects seen in *prg-1(-)* mutants (*Batista et al., 2008*; *Wang and Reinke, 2008*), we hypothesized that *snpc-1.3(-)* worms might also show fertility defects. Indeed, *snpc-1.3(-)* hermaphrodites exhibited significantly reduced fertility compared to wild type when grown at 25°C (*Figure 6A*). To address whether this decreased fertility was due to defects during spermatogenesis or oogenesis, we compared brood sizes from crosses of *fem-1(-)* females and *him-8(-)* males with or without *snpc-1.3*. Compared to *him-8(-)* males, *him-8(-); snpc-1.3(-)* males generated significantly smaller brood sizes when crossed with *fem-1(-)* females; in contrast, *fem-1(-); snpc-1.3(-)* and *fem-1(-)* females generated similar brood sizes when crossed with *him-8(-)* males (*Figure 6B*). As an orthogonal test, we crossed hermaphrodites to transgenic males expressing a fluorescent marker to facilitate counting of cross progeny. These transgenic males encode a reporter gene, *Pcol-19::gfp*, which drives GFP expression in the cuticle (*Figure 6—figure supplement 1A*). All *Pcol-19::gfp; snpc-1.3(-)* males produced fewer GFP+ progeny than wild-type *Pcol-19::gfp* males, whereas wild-type or *snpc-1.3(-)* hermaphrodites generated similar numbers of GFP+ progeny when crossed with wild-type *Pcol-19::gfp* males (*Figure 6—figure supplement 1A*). These results suggest that the reduced fertility of *snpc-1.3(-)* mutants likely reflect defects during spermatogenesis.

To investigate the cause of *snpc-1.3*-dependent loss of male fertility, we examined spermiogenesis and sperm morphology in *snpc-1.3(-)* males. After meiotic differentiation in the male germline, male spermatids are induced by ejaculation and undergo spermiogenesis, a process that converts immature spermatids to motile sperm with a functioning pseudopod. Spermiogenesis can be induced in vitro by isolating spermatids directly from males and treating them with pronase (*Shakes and Ward, 1989*). Males lacking *prg-1* still generate differentiated spermatids, but rarely produce normal pseudopodia upon activation (*Figure 6C,D*; *Wang and Reinke, 2008*). Similar to *prg-1(-)* mutants, *snpc-1.3(-)* spermatids were rarely able to form normal pseudopodia. In contrast, *snpc-1.3 (3xtbs)* sperm formed normal pseudopodia at a frequency similar to wild type (*Figure 6C, D*). In addition, many of the *snpc-1.3(-)* spermatids resembled sperm undergoing intermediate stages of spermiogenesis. Spermiogenesis, in vivo, starts off with spherical spermatids that enter into an intermediate stage characterized by the growth of spiky protrusions. This stage is then followed by fusion of the spiky protrusions into a motile pseudopod (*Figure 6E*). To understand the dynamics of *snpc-1.3(-)* sperm progression through spermiogenesis, we treated spermatids with pronase and observed each activated spermatid over time. Wild-type spermatids spent an average of 6.2 min ± 4.5 min in the intermediate state before polarization and pseudopod development. In contrast, *snpc-1.3(-)* spermatids occupied the intermediate state for a significantly shorter period of time (2.9 min ± 3.7 min, p<0.05; Student's t-test) before forming pseudopods. By tracking each individual spermatid across spermiogenesis, we found most *snpc-1.3(-)* spermatids were unable to sustain

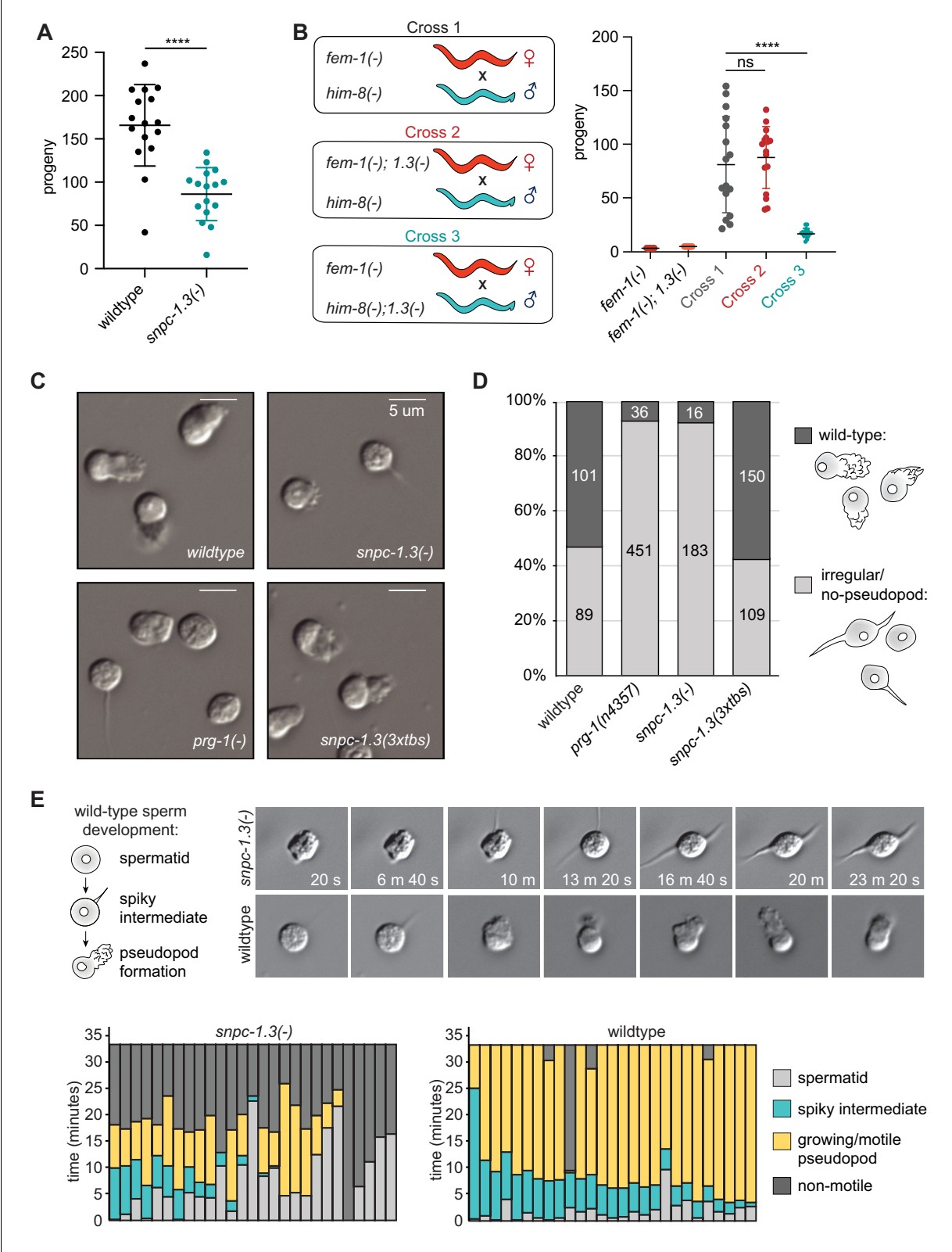

**Figure 6.** SNPC-1.3 is critical for male fertility. (**A**) *snpc-1.3(-)* hermaphrodites exhibit sterility at 25˚C. Circles correspond to the number of viable progeny from singled hermaphrodites (n = 16). Black bars indicate mean ± SD. Statistical significance was assessed using Welch's t-test (****p≤0.0001). (**B**) *snpc-1.3* promotes male fertility but is dispensable for female fertility. (Left) Diagram illustrates crosses between strains for mating assays (*1.3(-)* denotes *snpc-1.3(-)*). (Right) *snpc-1.3(-); him-8(-)* males crossed to *fem-1(-)* females show severe fertility defects (Cross 3). *snpc-1.3; fem-1(-)* females

*Figure 6 continued on next page*

*Figure 6 continued*

crossed to *him-8(-)* males (Cross 2) show equivalent fertility similar to *fem-1(-)* females crossed to *him-8(-)* males (Cross 1). Circles correspond to the number of viable progeny from cross (n = 16). Black bars indicate mean ± SD. Statistical significance was assessed using Welch's t-test (ns: not significant; ****p≤0.0001). (C) *snpc-1.3(-)* spermatids exhibit severe morphological defects. Images of pronase-treated sperm of wild-type, *prg-1(-)*, *snpc-1.3(-)*, and *snpc-1.3 (2xtbs)* males. (D) *snpc-1.3(-)* spermatids exhibit severe sperm maturation defects. (E) (Top) Images depicted at 3 min intervals of a sperm undergoing activation and maturation. Imaging of spermatid commenced ~3 min after pronase treatment. (Bottom) Graphical display of individual sperm tracked over time after pronase treatment.

The online version of this article includes the following source data and figure supplement(s) for figure 6:

**Source data 1.** Source data for *Figure 6A*.
**Source data 2.** Source data for *Figure 6B*.
**Source data 3.** Source data for *Figure 6E*.
**Figure supplement 1.** SNPC-1.3 is critical for male fertility.

pseudopod growth. While wild-type spermatids exhibited pseudopod growth and motility for an average of 24 min ± 10.35 min, *snpc-1.3(-)* spermatids sustained growth for a significantly shorter period of time (7.3 min ± 5.7 min, p<0.05; Student's t-test) before becoming immotile (*Figure 6E*). These results indicate that *spnc-1.3(-)* males have defective spermatogenesis processes and exhibit similar fertility defects as *prg-1(-)* mutants.

## Discussion

Our data indicate that *C. elegans* SNPC-1.3, a human SNAPC1 ortholog, functions as a male piRNA transcription factor. SNPC-1.3 interacts with SNPC-4 in foci in male germ cell nuclei (*Figure 2*) and, by preferentially binding male piRNA promoters (*Figure 4*), is critical for their expression (*Figure 3*). SNPC-1.3 expression, reflecting the developmental profile of male piRNAs (*Figure 5—figure supplement 1A*), is highest during spermatogenesis. We demonstrate that the *snpc-1.3* locus itself is regulated by the sex determination regulator, TRA-1 (*Figure 5*). During spermatogenesis, *tra-1* expression is low, and *snpc-1.3* and other male-promoting genes are licensed for expression. In contrast, *tra-1* expression is upregulated during oogenesis and TRA-1 binds the *snpc-1.3* promoter to repress its transcription, leading to the expression of female over male piRNAs (*Figure 7*). We propose that SNPC-1.3, via its interaction with SNPC-4, can direct the specificity of the core piRNA complex preferentially to male piRNA loci.

### How is the expression of male and female piRNAs coordinated?

Given its role as a putative male piRNA transcription factor, we expected that deletion of *snpc-1.3* would result in loss of male piRNAs with no consequences to the expression of female piRNAs. However, loss of *snpc-1.3* also results in increased female piRNA expression during spermatogenesis (*Figure 3*), whereas ectopic overexpression of *snpc-1.3* during oogenesis leads to decreased female piRNA levels (*Figure 5*). Taken together, our findings suggest that transcription of male and female piRNAs is not completely separable from each other and that the balance in expression of the two piRNA subclasses may be dictated by the allocation of shared core transcription factors such as SNPC-4.

Similar to multiple gene classes activated by general transcription factors (*Levine et al., 2014*), we speculate that male and female promoters compete for access to a limited pool of the core transcription complex, which includes SNPC-4, PRDE-1, TOFU-4, and TOFU-5 (*Figure 1*). Therefore, we propose a model in which the expression and binding of SNPC-1.3 to core piRNA factors serves to 'sequester' the core complex away from female promoters. Mechanistically, we posit that the core piRNA transcription complex is specified to female promoters, and that only upon association with SNPC-1.3 is the core machinery directed to male promoters. We predict that when SNPC-1.3 is absent, more SNPC-4 and other previously identified cofactors are available to transcribe female piRNAs. Conversely, overexpression of SNPC-1.3 leads to the disproportionate recruitment of the core machinery to male promoters, leading to the indirect downregulation of female piRNAs. By controlling male piRNA expression, SNPC-1.3 is crucial for maintaining the balance between male and female piRNA levels across development.

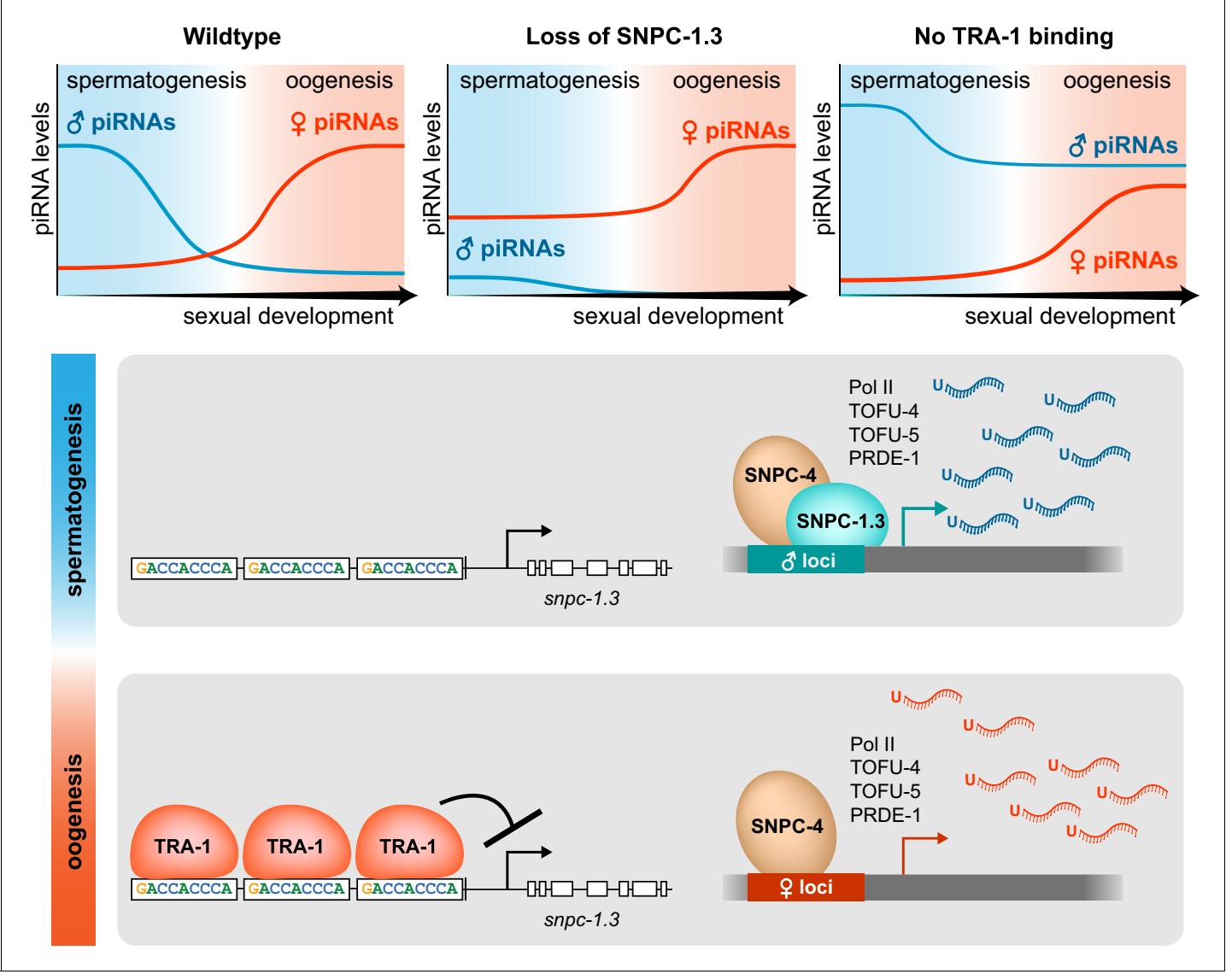

**Figure 7.** Model illustrating the dynamics of male and female piRNA transcription across *C. elegans* sexual development. In wild-type worms, male and female piRNA expression peaks during spermatogenesis and oogenesis, respectively. (Top) In *snpc-1.3(-)* mutants, male piRNA expression is abrogated, and female piRNA expression is moderately enhanced across sexual development relative to wild type. In TRA-1 binding site mutants, *snpc-1.3* expression is de-repressed causing ectopic upregulation of male piRNAs and moderate repression of female piRNA expression during oogenesis relative to wild type. (Bottom) During spermatogenesis, SNPC-1.3 interacts with SNPC-4 at male promoters to drive male piRNA transcription. During oogenesis, TRA-1 represses the transcription of *snpc-1.3* which results in the suppression of male piRNA transcription, thus leading to enhanced transcription of female piRNAs.

While the default specification of the core complex to female promoters presents perhaps the most parsimonious explanation underlying male and female piRNA expression, we cannot exclude the possibility that an additional female-specific *trans*-acting factor may direct the core piRNA complex to female promoters. If true, we speculate that the developmental expression of such a factor (low during spermatogenesis and high during oogenesis), coupled with the developmental expression of SNPC-1.3, would coordinate the differential expression of male and female piRNAs. During spermatogenesis, SNPC-1.3 is more highly expressed such that the core machinery would primarily be directed to male promoters. In contrast, during oogenesis, SNPC-1.3 expression is low, concomitant with elevated expression of a female factor to license transcription of female piRNAs. This model, where both factors are present during both spermatogenesis and oogenesis, but in different

ratios, would also be consistent with our piRNA expression analysis in *snpc-1.3* loss-of-function and overexpression mutants.

## The piRNA pathway co-opts snRNA biogenesis machinery

Our work adds to a growing body of evidence that snRNA machinery has been hijacked at multiple stages in *C. elegans* piRNA biogenesis, including transcription (*Kasper et al., 2014*; *Weng et al., 2019*) and termination (*Beltran et al., 2019*). Investigating potential parallels between snRNA and piRNA biogenesis may provide useful clues into the role of SNPC-1.3 in the piRNA complex.

The minimal snRNA SNAP complex consists of a 1:1:1 heterotrimer of the subunits SNAPC4, SNAPC1, and SNAPC3 in humans and SNAP190, SNAP43, and SNAP50 in flies (*Henry et al., 1998*; *Hung and Stumph, 2011*; *Li et al., 2004*; *Ma and Hernandez, 2002*; *Mittal et al., 1999*; *Figure 1*). In vitro studies have shown that the trimer must assemble before the complex is able to bind DNA. Similarly, our data show SNPC-1.3 requires SNPC-4 to bind at the piRNA clusters (*Figure 4*), although we cannot formally rule out that loss of SNPC-4 only affects the stability of SNPC-1.3, rather than directly recruiting SNPC-1.3 to piRNA promoters. We speculate the piRNA complex is assembled in a similar fashion to the snRNA complex. Based on this model, we expect that SNPC-4 binding at male piRNA loci is abolished in a *snpc-1.3* mutant. However, conclusive evidence that SNPC-4 binding at male piRNA promoters requires SNPC-1.3 is still lacking. Due to the highly clustered nature of *C. elegans* piRNAs, we anticipate that detecting differences in SNPC-4 binding between male and female piRNAs in *snpc-1.3(-)* mutants may not be possible with traditional ChIP-seq methods, and may require application of higher resolution techniques.

Given that piRNAs have co-opted *trans*-acting factors from snRNA biogenesis (*Kasper et al., 2014*), it would not be surprising if piRNAs also co-evolved *cis*-regulatory elements for transcription factor binding from snRNA loci. Recently, *Beltran et al., 2019* identified similarity between the 3′ end of PSEs of snRNA promoters and the eight nt piRNA core motif in nematodes. In addition, Pol II and Pol III transcription from snRNA promoters share a common PSE, but are distinguished by the presence of other unique motifs (*Hung and Stumph, 2011*). Correspondingly, the canonical Type I and less abundant Type II piRNAs can be discriminated by the presence or absence of the eight nt core motif, respectively. Factors such as TOFU-4 and TOFU-5 function in both Type I and II piRNA expression, whereas PRDE-1 is only required for Type I piRNAs (*Kasper et al., 2014*; *Weng et al., 2019*). Altogether, these observations highlight the importance of *cis*-regulatory elements in specifying the expression of snRNAs and piRNA classes. In addition to enrichment of cytosine at the 5′ position in the male core motif (*Billi et al., 2013*), we hypothesize that as-yet unidentified motifs may further discriminate male from female piRNA promoters. While we observed SNPC-1.3 binding to be enriched upstream of male piRNA loci (*Figure 4*), we cannot definitively conclude that SNPC-1.3 binds to the male-specific core motif, given the limitations of conventional ChIP-seq in resolving the SNPC-1.3 footprint. Identifying the factors that specifically bind the eight nt core motif and other potential *cis*-regulatory elements important for sex-specific piRNA expression will require further investigation.

## What are the functions of male piRNAs in *C. elegans*?

Our data suggest that SNPC-1.3 is essential for proper spermiogenesis (*Figure 6*). We hypothesize the global loss of male piRNAs in a *snpc-1.3(-)* mutant is responsible for the higher incidence of spermiogenesis arrest and subsequent loss in fertility, although it is possible that SNPC-1.3 may have other or additional effects on male fertility. Characterization of *prg-1(-)* mutants during spermiogenesis agree with our findings that loss of piRNAs in the male germline leads to acute defects directly responsible for fertility (*Wang and Reinke, 2008*). Since the initial discovery of piRNA function in the targeting and silencing of transposons in *Drosophila* (*Vagin et al., 2006*; *Brennecke et al., 2007*), analyses in other systems have revealed that piRNAs have acquired neofunctions at later points along the evolutionary time scale (*Ozata et al., 2019*).

While it is estimated that as much as 45% of the human genome encodes for transposable elements (*Lander et al., 2001*), only 12% of *C. elegans* genome encodes such elements. Furthermore, nearly all of these regions are inactive in *C. elegans* (*Bessereau, 2006*). In contrast to *Drosophila* piRNAs that target and silence transposons with perfect complementarity (*Brennecke et al., 2007*), *C. elegans* piRNAs are thought to bind a broad range of endogenously expressed transcripts by

partial complementarity (*Ashe et al., 2012*; *Shen et al., 2018*; *Zhang et al., 2018*). Together, these findings suggest that worm piRNAs function in capacities distinct from canonical transposon silencing. While a recent methodology used crosslinking, ligation, and sequencing of piRNA:target hybrids (CLASH) to determine that female piRNAs engage with almost every germline transcript (*Shen et al., 2018*), how female piRNAs select their targets has yet to be examined. Like piRNAs characterized in the female germline, male piRNAs may be interfacing with a broad range of targets to regulate gene expression for proper spermatogenesis. Loss of *prg-1* in males causes the downregulation of a subset of spermatogenesis-specific genes (*Wang and Reinke, 2008*), suggesting male piRNAs serve a protective function for spermatogenic processes. The characterization of the in vivo landscape of male piRNA target selection using CLASH may provide insights into piRNA function during spermatogenesis.

## Why are male piRNAs restricted from the female germline?

Sperm and oocytes pass epigenetic information such as noncoding RNAs to the next generation (*Hammoud et al., 2014*; *Brykczynska et al., 2010*; *Tabuchi et al., 2018*; *Kaneshiro et al., 2019*). Recent studies show maternal piRNAs trigger the production of endo-siRNAs, called 22G-RNAs for their 5′ bias for guanine and 22 nt length, to transmit an epigenetic memory of foreign versus endogenous elements to the next generation (*Ashe et al., 2012*; *Buckley et al., 2012*; *Shirayama et al., 2012*). We predict that misexpression of male piRNAs during oogenesis may perturb the native pool of female piRNAs necessary for appropriate recognition of self versus non-self elements. This may explain the decrease in fertility we observed in multiple TRA-1 binding site mutant hermaphrodites (*Figure 6—figure supplement 1B*). As *snpc-1.3 (3xtbs)* sperm do not seem to exhibit significant morphological defects (*Figure 6*), the fertility defects in the *snpc-1.3 (3xtbs)* mutants could be due to problems arising in oogenesis. However, based on our sequencing data in *snpc-1.3 (2xtbs)* mutants, we cannot distinguish whether fertility defects during oogenesis are due to upregulation of male piRNAs, downregulation of female piRNAs, a combination of the two, or misexpression of downstream endo-siRNAs triggered by piRNAs. Further study of *snpc-1.3* gain-of-function mutants in oogenesis will enhance our understanding of the physiological consequences of expressing male piRNAs in the female germline.

## The intersection between sex determination and sex specification of piRNA expression

We speculate that gene duplication of the *snpc-1* family of genes occurred early during nematode evolution and allowed for the acquisition of new functions by *snpc-1* paralogs, specifically, from snRNA to piRNA biogenesis. At least two SNPC-1 paralogs are present within the distantly related nematode species, *Plectus sambesii*. Furthermore, we predict that co-opting SNPC-1 paralogs for piRNA biogenesis may have occurred in parallel with the evolution of the nematode sex determination pathway. TRA-1 is a sex determination factor that acts to repress male-promoting gene expression in female germ cells to promote female germ cell fate. While *Drosophila* sex determination utilizes different factors than *C. elegans*, further investigation into the conservation of TRA-1 shows that it is a common feature in at least the nematode lineage (*Pires-daSilva and Sommer, 2004*). Additionally, just as we have shown that TRA-1 represses *snpc-1.3* in *C. elegans* (*Figure 5*), TRA-1 binding motifs GGG(A/T)GG are present in the putative upstream promoter regions of *snpc-1.3* homologs identified in *C. briggsae*, *C. brenneri*, and *C. nigoni* (*Figure 6—figure supplement 1C*). Taken together, these analyses point to a conserved link between sex determination and piRNA biogenesis pathways among nematodes.

In summary, our work reveals that SNPC-1.3 is specified to the male germline and is essential for male piRNA expression. We have identified SNPC-1.3 as a major target of TRA-1 repression in the female germline. Future studies will likely uncover additional factors required to coordinate the proper balance of sex-specific piRNAs required for proper germline development and animal fertility.

### Contact for reagent and resource sharing

More details about resources and reagents can be found in the Key Resources Table found in Materials and methods. Further information and requests for resources and reagents should be directed to and will be fulfilled by the Lead Contact, John K. Kim (jnkim@jhu.edu).

### Experimental model and subject details

*C. elegans* strains were maintained at 20°C according to standard procedures (*Brenner, 1974*), unless otherwise stated. Bristol N2 was used as the wild-type strain. Except for RNAi and ChIP experiments, worms were fed *E. coli* strain OP50. Worms used for ChIP were fed *E. coli* strain HB101.

## Materials and methods

**Key resources table**

| Reagent type (species) or resource | Designation | Source or reference | Identifiers | Additional information |
|---|---|---|---|---|
| Antibody | Mouse monoclonal anti-Flag | Sigma | F1804; RRID: AB_262044 | Western – 1:1000; IF – 1:200 |
| Antibody | Rabbit polyclonal anti-H3 | Abcam | Ab12079; RRID: AB_298834 | 1:15,000 |
| Antibody | Rabbit polyclonal anti-tubulin | Sigma–Aldrich | T1450; RRID:AB_261655 | 1:5000 |
| Antibody | Goat polyclonal anti-rabbit | Jackson Laboratories | 111035045; RRID:AB_2337938 | 1:15,000 |
| Antibody | Sheep polyclonal anti-mouse | GE Healthcare | NA931; RRID:AB_772210 | 1:5000 |
| Antibody | Rat monoclonal anti-Ollas | Novus Biologicals | NBP1-06713SS | Western – 1:8,000; IF – 1:200 |
| Antibody | Polyclonal donkey anti-rabbit, AlexaFluor 488 | ThermoFisher | A-21208 | 1:400 |
| Antibody | Polyclonal goat anti-mouse, AlexaFluor 555 | ThermoFisher | A-21127 | 1:400 |
| Antibody | Polyclonal goat anti-mouse IgG (IRDye 800 CW) | LI-COR Biosciences | 925–32210; RRID:AB_621842 | 1:15,000 |
| Antibody | Polyclonal goat anti-rabbit IgG (IRDye 680 RD) | LI-COR Biosciences | 925–68071; RRID:AB_2721181 | 1:15,000 |
| Other | DAPI | ThermoFisher | 62248 | 0.5 µg/mL |
| Other | Vectashield with DAPI | Vector Laboratories | H-1200; RRID:AB_2336790 | |
| Other | Roche Blocking Buffer | Millipore Sigma | 11096176001 | |
| Other | Odyssey Blocking Buffer (TBS) | LI-COR Biosciences | 927–50003 | |
| Strain, strain background (*E. coli*) | OP50 | Shared Fermentation Facility, The Pennsylvania State University | | |

*Continued on next page*

*Continued*

| Reagent type (species) or resource | Designation | Source or reference | Identifiers | Additional information |
|---|---|---|---|---|
| Strain, strain background (*E. coli*) | HB101 | Shared Fermentation Facility, The Pennsylvania State University | | |
| Strain, strain background (*E. coli*) | HT115 RNAi clones | *Kamath and Ahringer, 2003* | | |
| Other | TriReagent | ThermoFisher | AM9738 | |
| Other | Benzonase | Sigma–Aldrich | E1014 | 1:1000 |
| Other | RNA 5′ Polyphosphatase | Illumina | RP8092H | |
| Other | Multiscribe Reverse Transcriptase | ThermoFisher | 4311235 | |
| Other | Absolute Blue SYBR Green | ThermoFisher | AB4166B | |
| Other | Dimethyl pimelimidate dihydrochloride | Sigma–Aldrich | D8388 | |
| Other | Protease inhibitor cocktail | Roche | 4693159001 | 1:100 |
| Other | Purelink RNAse A | ThermoFisher | 12091021 | 1:10 |
| Other | Pronase E | Sigma–Aldrich | 7433–2 | 20 µg/mL |
| Other | TaqMan Universal PCR Master Mix, No AmpErase UNG | ThermoFisher | 4324018 | |
| Sequence-based reagent | U18 TaqMan probe | ThermoFisher | 1764 | TGGCAGTGATGATCACAAATCCGTGTTTCTGA CAAGCGATTGACGATAGAAAACCGGCTGAGCCA |
| Sequence-based reagent | 21UR-1848 TaqMan probe | ThermoFisher | | UAAAGGCAGAAUUUUAUCAAC |
| Sequence-based reagent | 21UR-2502 TaqMan probe | ThermoFisher | | UGAAAUUGUAGUAGACUGCUG |
| Sequence-based reagent | 21UR-4807 TaqMan probe | ThermoFisher | | UGGGUGAAUUCUGUCCCGAAC |
| Sequence-based reagent | 21UR-1258 TaqMan probe | ThermoFisher | | UAGACUUGAGUUAGAACGGUU |
| Sequence-based reagent | 21UR-3142 TaqMan probe | ThermoFisher | | GUAGGGUCGUCUCUUGAGAGC |
| Sequence-based reagent | 21UR-3766 TaqMan probe | ThermoFisher | | UGGAAGCUUGAUGGAAAAUGC |
| Commercial assay kit | NEBNext Multiplex Small RNA Library Prep Set for Illumina | New England Biolabs | E7330S | |
| Other | Small RNA-seq data | This study | GEO: GSE152831 | |
| Other | ChIP-seq data | This study | GEO: GSE152831 | |
| Other | Mass spec data | This study | GEO: GSE152831 | |
| Strain, strain background (*C. elegans*) | wild-type, Bristol isolate | CGC | N2 | |
| Strain, strain background (*C. elegans*) | *prg-1(n4357) I* | CGC | SX922 | |

*Continued on next page*

*Continued*

| Reagent type (species) or resource | Designation | Source or reference | Identifiers | Additional information |
|---|---|---|---|---|
| Strain, strain background (*C. elegans*) | *snpc-1.3(xk27)[snpc-1.3(lof)] V* | This study | QK171 | For CRISPR/Cas9 reagents and methodology, see *Supplementary file 4* and Method details. |
| Strain, strain background (*C. elegans*) | *fem-1(hc17) IV* | CGC | BA17 | |
| Strain, strain background (*C. elegans*) | *him-8(e1489) IV* | CGC | CB1489 | |
| Strain, strain background (*C. elegans*) | *snpc-1.3(xk28)[snpc-1.3 (1xtbs)] V* | This study | QK172 | For CRISPR/Cas9 reagents and methodology, see *Supplementary file 4* and Method details. |
| Strain, strain background (*C. elegans*) | *snpc-1.3(xk29)[snpc-1.3 (2xtbs)] V* | This study | QK173 | For CRISPR/Cas9 reagents and methodology, see *Supplementary file 4* and Method details. |
| Strain, strain background (*C. elegans*) | *snpc-1.3(xk30)[snpc-1.3 (3xtbs)] V* | This study | QK174 | For CRISPR/Cas9 reagents and methodology, see *Supplementary file 4* and Method details. |
| Strain, strain background (*C. elegans*) | *snpc-4(xk31)[snpc-4::3xflag] I* | This study | QK175 | For CRISPR/Cas9 reagents and methodology, see *Supplementary file 4* and Method details. |
| Strain, strain background (*C. elegans*) | *snpc-4(xk31)[snpc-4::3xflag] I; fem-1 (hc17) IV* | This study | QK176 | For CRISPR/Cas9 reagents and methodology, see *Supplementary file 4* and Method details. |
| Strain, strain background (*C. elegans*) | *snpc-4(xk31)[snpc-4::3xflag] I; him-8 (e1489) IV* | This study | QK177 | For CRISPR/Cas9 reagents and methodology, see *Supplementary file 4* and Method details. |
| Strain, strain background (*C. elegans*) | *snpc-1.3(xk27)[snpc-1.3(lof)] V; fem-1 (hc17) IV* | This study | QK178 | For CRISPR/Cas9 reagents and methodology, see *Supplementary file 4* and Method details. |
| Strain, strain background (*C. elegans*) | *snpc-1.3(xk27)[snpc-1.3(lof)] V; him-8 (e1489) IV* | This study | QK179 | For CRISPR/Cas9 reagents and methodology, see *Supplementary file 4* and Method details. |
| Strain, strain background (*C. elegans*) | *mals105[col-19:: GFP] V* | Xantha Karp lab | XV33 | |
| Strain, strain background (*C. elegans*) | *snpc-1.3(xk27)[snpc-1.3(lof)] V; mals105 V* | This study | QK180 | For CRISPR/Cas9 reagents and methodology, see *Supplementary file 4* and Method details. |
| Strain, strain background (*C. elegans*) | *snpc-4(xk31)[snpc-4::3xflag] I, snpc-1.3 (xk27)[snpc-1.3(lof)] V* | This study | QK181 | For CRISPR/Cas9 reagents and methodology, see *Supplementary file 4* and Method details. |
| Strain, strain background (*C. elegans*) | *snpc-4(xk23) I [snpc-4::aid::ollas]* | This study | QK162 | For CRISPR/Cas9 reagents and methodology, see *Supplementary file 4* and Method details. |
| Strain, strain background (*C. elegans*) | *snpc-4(xk23) I; unc-11(ed3) III; ieSi38 IV* | This study | QK163 | For CRISPR/Cas9 reagents and methodology, see *Supplementary file 4* and Method details. |
| Strain, strain background (*C. elegans*) | *unc-11(ed3) III; ieSi38 IV [sun-1p:: TIR1::mRuby::sun-1 3' UTR + Crb-unc-119 (+)] IV* | CGC | CA1199 | |

*Continued on next page*

*Continued*

| Reagent type (species) or resource | Designation | Source or reference | Identifiers | Additional information |
|---|---|---|---|---|
| Strain, strain background (*C. elegans*) | *glp-4(bn2) I* | CGC | SS104 | |
| Strain, strain background (*C. elegans*) | *snpc-1.3(xk32)[snpc-1.3a::3xflag] V* | This study | QK182 | For CRISPR/Cas9 reagents and methodology, see *Supplementary file 4* and Method details. |
| Strain, strain background (*C. elegans*) | *glp-4(bn2) I; snpc-1.3(xk32)[snpc-1.3a::3xflag] V* | This study | QK183 | For CRISPR/Cas9 reagents and methodology, see *Supplementary file 4* and Method details. |
| Strain, strain background (*C. elegans*) | *snpc-1.3(xk33)[snpc-1.3a::ollas] V* | This study | QK184 | For CRISPR/Cas9 reagents and methodology, see *Supplementary file 4* and Method details. |
| Strain, strain background (*C. elegans*) | *snpc-4(xk31)[snpc-4::3xflag] I, snpc-1.3(xk33)[snpc-1.3a::ollas] V* | This study | QK185 | For CRISPR/Cas9 reagents and methodology, see *Supplementary file 4* and Method details. |
| Strain, strain background (*C. elegans*) | *prde-1(xk34)[prde-1::ollas] V* | This study | QK186 | For CRISPR/Cas9 reagents and methodology, see *Supplementary file 4* and Method details. |
| Strain, strain background (*C. elegans*) | *snpc-4(xk31)[snpc-4::3xflag] I; prde-1(xk34)[prde-1::ollas] V* | This study | QK187 | For CRISPR/Cas9 reagents and methodology, see *Supplementary file 4* and Method details. |
| Strain, strain background (*C. elegans*) | *snpc-4(xk35)[snpc-4::ollas] I* | This study | QK188 | For CRISPR/Cas9 reagents and methodology, see *Supplementary file 4* and Method details. |
| Strain, strain background (*C. elegans*) | *snpc-4(xk35)[snpc-4::ollas] I; snpc-1.3(xk32)[snpc-1.3a::3xflag] V* | This study | QK189 | For CRISPR/Cas9 reagents and methodology, see *Supplementary file 4* and Method details. |
| Strain, strain background (*C. elegans*) | *snpc-4(xk31)[snpc-4::3xflag] I; snpc-1.3(xk29)[snpc-1.3(2xtbs)] V* | This study | QK190 | For CRISPR/Cas9 reagents and methodology, see *Supplementary file 4* and Method details. |
| Strain, strain background (*C. elegans*) | *snpc-4(xk23) I; unc-11(ed3) III; ieSi38 IV; snpc-1.3(xk32)[snpc-1.3a::3xflag] V* | This study | QK191 | For CRISPR/Cas9 reagents and methodology, see *Supplementary file 4* and Method details. |
| Strain, strain background (*C. elegans*) | *snpc-1.3(xk36)[snpc-1.3a::3xflag(2xtbs)] V* | This study | QK192 | For CRISPR/Cas9 reagents and methodology, see *Supplementary file 4* and Method details. |
| Strain, strain background (*C. elegans*) | *tra-1(xk37)[3xflag::tra-1] III* | This study | QK193 | For CRISPR/Cas9 reagents and methodology, see *Supplementary file 4* and Method details. |
| Strain, strain background (*C. elegans*) | *tra-1(xk37)[3xflag::tra-1] III; snpc-1.3(xk28)[snpc-1.3(1xtbs)] V* | This study | QK194 | For CRISPR/Cas9 reagents and methodology, see *Supplementary file 4* and Method details. |
| Strain, strain background (*C. elegans*) | *tra-1(xk37)[3xflag::tra-1] III; snpc-1.3(xk29)[snpc-1.3(2xtbs)] V* | This study | QK195 | For CRISPR/Cas9 reagents and methodology, see *Supplementary file 4* and Method details. |
| Strain, strain background (*C. elegans*) | *tra-1(xk37)[3xflag::tra-1] III; snpc-1.3(xk30)[snpc-1.3(3xtbs)] V* | This study | QK196 | For CRISPR/Cas9 reagents and methodology, see *Supplementary file 4* and Method details. |

*Continued on next page*

*Continued*

| Reagent type (species) or resource | Designation | Source or reference | Identifiers | Additional information |
|---|---|---|---|---|
| Strain, strain background (*C. elegans*) | *snpc-1.3(xk38) [snpc-1.3b::3xflag] V* | This study | QK197 | For CRISPR/Cas9 reagents and methodology, see *Supplementary file 4* and Method details. |
| Software, algorithm | bbmap 38.23 | http://jgi.doe.gov/data-and-tools/bb-tools | | |
| Software, algorithm | Bowtie 1.1.1 | *Langmead et al., 2009* | | |
| Software, algorithm | Bowtie2 2.3.4.2 | *Langmead and Salzberg, 2012* | | |
| Software, algorithm | CASHX 2.3 | *Fahlgren et al., 2009* | | |
| Software, algorithm | deepTools 3.3.1 | *Ramírez et al., 2016* | | |
| Software, algorithm | DESeq2 1.18.1 | *Love et al., 2014* | | |
| Software, algorithm | DESeq2 1.26.0 | *Love et al., 2014* | | |
| Software, algorithm | FastQC 0.11.7 | http://www.bioinformatics.babraham.ac.uk/projects/fastqc/ | | |
| Software, algorithm | GraphPad Prism | https://www.graphpad.com | | |
| Software, algorithm | ImageJ | ImageJ | | |
| Software, algorithm | MACS 2.1.2 | *Zhang et al., 2008* | | |
| Software, algorithm | MEME suite 5.1.1 | *Bailey et al., 2009* | | |
| Software, algorithm | RStudio 3.4.1 | https://www.rstudio.com | | |
| Software, algorithm | Samtools 1.9 | *Li et al., 2009* | | |
| Software, algorithm | Subread 1.6.3 | *Liao et al., 2014* | | |
| Software, algorithm | Trim Galore! 0.5.0 | http://www.bioinformatics.babraham.ac.uk/projects/trim_galore/ | | |
| Software, algorithm | Trimmomatic 0.39 | *Bolger et al., 2014* | | |
| Other | Dynabeads Protein G | ThermoFisher | 10004D | |
| Other | Dynabeads M280 sheep anti-mouse IgG | ThermoFisher | 11202D | |
| Other | SuperScript III Reverse Transcriptase | ThermoFisher | 18080085 | |

## Generations of strains

CRISPR/Cas9-generated strains were created as described in *Paix et al., 2015* and are listed in *Supplementary file 4*. crRNA and repair template sequences of CRISPR-generated strains are listed in *Supplementary file 4*. After initial phenotyping of *snpc-1.3a::3xflag* and *snpc-1.3b::3xflag* (*Figure 1—figure supplement 1C−G*), *snpc-1.3a::3xflag* was used for all subsequent experiments (and is referred to as *snpc-1.3::3xflag*).

## RNAi assays

Bacterial RNAi clones were grown from the Ahringer RNAi library (*Kamath and Ahringer, 2003*). Synchronized L1 worms were plated on HT115 bacteria expressing dsRNA targeting the gene interest or L4440 empty vector as a negative control as previously described (*Timmons and Fire, 1998*). All RNAi experiments were performed at 20°C unless otherwise stated.

## RNA extraction, library preparation, and sequencing

After hypochlorite preparation and hatching in M9 buffer, *snpc-4::aid::ollas* and *snpc-4::aid::ollas; Psun-1::TIR1* worms were transferred from NGM plates to plates containing 250 µM auxin 20 hr before collecting L4 and gravid worms, 48 hr and 72 hr after plating L1 worms at 20°C, respectively. Worms were collected in TriReagent (ThermoFisher Scientific) and subjected to three freeze–thaw cycles. Following addition of 1-bromo-3-chloropropane, the aqueous phase was then precipitated with isopropanol at −80°C for 2 hr. To pellet RNA, samples were spun at 21,000 × $g$ for 30 min at 4°C. After three washes in 75% ethanol, the pellet was resuspended in water.

RNA concentration and quality were measured using a TapeStation (Agilent Technologies). We size-selected small RNAs of 16-30 nt in length from 5 µg total RNA on 17% denaturing polyacrylamide gels. Small RNAs were treated with 5′ polyphosphatase (Illumina) to reduce 5′ triphosphate groups to monophosphates to enable 5′ adapter ligation. Small RNA-sequencing libraries were prepared using the NEBNext Multiplex Small RNA Library Prep Set for Illumina (NEB). Small RNA amplicons were size-selected on 10% polyacrylamide gels and quantified using qRT-PCR. Samples for each developmental time point were pooled into a single flow cell and single-end, 75 nt reads were generated on a NextSeq 500 (Illumina). An average of 42.01 million reads (range 33.05–50.39 million) was obtained for each library.

## Quantitative RT-PCR

Taqman cDNA synthesis was performed as previously described (*Weiser et al., 2017*). Briefly, for quantification of piRNA levels, TaqMan small RNA probes were designed and synthesized by Applied Biosystems. All piRNA species assessed by qPCR were normalized to U18 small nucleolar RNA. Fifty nanograms of total RNA was used for cDNA synthesis. cDNA was synthesized by Multiscribe Reverse Transcriptase (Applied Biosystems) using the Eppendorf Mastercycler Pro S6325 (Eppendorf). Detection of small RNAs was performed using the TaqMan Universal PCR Master Mix and No AmpErase UNG (Applied Biosystems). For quantification of mRNA levels, cDNA was made using 500 ng of total RNA using Multiscribe Reverse Transcriptase (Applied Biosystems). For quantification of snRNA levels, cDNA was made using 250 ng of total RNA using SuperScript III Reverse Transcriptase (ThermoFisher). Assays for mRNA and snRNA levels were performed with Absolute Blue SYBR Green (ThermoFisher) and normalized to *eft-2* using CFX63 Real Time System Thermocyclers (Bio-Rad). All qPCR primers used are listed in *Supplementary file 4*.

## Covalent crosslinking of Dynabeads

Protein G Dynabeads (ThermoFisher Scientific, 1003D) were coupled to monoclonal mouse anti-FLAG antibody M2 (Sigma–Aldrich, F1804). After three washes in 1× PBST (0.1% Tween), Dynabeads were resuspended with 1× PBST with antibody, for a final concentration of 50 µg antibody per 100 µL beads. The antibody-bead mixture was nutated for 1 hr at room temperature. After three washes in 1× PBST and two washes in 0.2 M sodium borate pH 9.0, beads were nutated in 22 mM DMP (Sigma–Aldrich, D8388) in 0.2 M sodium borate for 30 min at room temperature. Following two washes in ethanolamine buffer (0.2 M ethanolamine, 0.2 M NaCl pH 8.5), beads were nutated for 1 hr at room temperature in the same buffer. Beads were placed into the same volume of ethanolamine buffer as the starting bead volume for storage at 4°C until use.

## Immunoprecipitation for mass spectrometry, co-IP experiments, and expression

For SNPC-4 IP mass spectrometry, synchronized populations of ~200,000,000 *him-8(e1489)* L4s and ~50,000,000 *fem-1(hc17)* females were grown at 25°C and collected on OP50. For co-IP experiments, ~500,000 L4 and ~250,000,000 gravid worms were grown and collected from OP50 plates. Due to low expression of SNPC-1.3 and appearance of background bands, samples examining SNPC-1.3 expression were subjected to immunoprecipitation before western blotting. For *glp-4 (bn2)*, *him-8(e1489)*, and *fem-1(hc17)* temperature-shift experiments, worms were grown at 15°C before hypochlorite treatment to isolate embryos. Synchronized L1s were then transferred to 25°C. For SNPC-1.3 expression in males and females, *snpc-1.3::3xflag; him-8(e1489)* L4 worms and *snpc-1.3::3xflag; fem-1(hc17)* adult worms were collected.

Unless otherwise stated, all samples for mass spectrometry, co-IP, and western blotting used in this study were subjected to the following procedure. After three washes in M9 and one wash in water, worms were frozen and ground using the Retsch MM400 ball mill homogenizer for two rounds of 1 min at 30 Hz. Frozen worm powder was resuspended in 1× lysis buffer used previously (*Moissiard et al., 2014*, 50 mM Tris–HCl pH 8.0, 150 mM NaCl, 5 mM MgCl$_2$, 1 mM EGTA, 0.1% NP-40, 10% glycerol) and protease inhibitor cocktail (Roche). After Bradford assay (Thermo-Fisher Scientific), lysates were normalized using lysis buffer and protease inhibitor. Benzonase (Sigma–Aldrich, E1014) was added to a final concentration of 1 µL/mL of lysate and nutated for 10 min at 4°C. After centrifugation for 10 min at 4,000 x *g*, 1 mL of supernatant was added to 50 µL of crosslinked Dynabeads and nutated for 15 min at 4°C. Samples were then washed three times in 1× lysis buffer with protease inhibitors before 1 hr nutation in 50 µL of 2 mg/mL FLAG peptide (Sigma–Aldrich, F4799) diluted in 1× lysis buffer. Complete eluate, as well as 5% of crude lysate (after addition of benzonase), input, pellet, and post-IP samples, were added to 2× Novex Tris–glycine sodium dodecyl sulfate sample buffer (ThermoFisher Scientific, LC2676) to 1×. Samples were then subjected to western blotting as described below.

## Western blotting

Co-IP samples and SNPC-1.3::3xFlag westerns in *snpc-1.3 tbs* mutants were run on either 8–16% or 8% Novex WedgeWell Tris–glycine precast gels (ThermoFisher) and transferred to PVDF membrane (Millipore). Mouse anti-Flag, rat anti-Ollas, rabbit anti-gamma tubulin, and rabbit anti-H3 were used at 1:1000, 1:8000, 1:5000, and 1:15000, respectively. Anti-mouse and anti-rabbit (for tubulin) antibodies were used at 1:5,000. To blot for H3, anti-rabbit secondary was used at 1:15,000. Anti-rat antibodies were used at 1:8,000. Antibodies used were Sigma–Aldrich F1804 (mouse anti-Flag), Novus Biologicals NBP1-06713SS (rat anti-Ollas), Sigma–Aldrich T1450 (rabbit anti-gamma tubulin), Abcam ab1791 (rabbit anti-H3), GE Healthcare NA931 (sheep anti-mouse), and Jackson Laboratories 111035045 (goat anti-rabbit). Both high-sensitivity Amersham ECL Prime (GE Healthcare, RPN2232) (for SNPC-1.3 blotting) and regular sensitivity Pierce ECL (ThermoFisher, 32209) were used for exposure in a Bio-Rad ChemiDoc Touch system.

For measuring SNPC-1.3 expression levels in various backgrounds, input (for normalization) and immunoprecipitation samples were run on 10% Novex WedgeWell Tris–glycine precast gels (ThermoFisher). Following transfer, the membrane was dried for 20 min at room temperature. The blot was then recharged in 100% methanol for 1 min, followed by a water rinse and a wash in TBS for 2 min. Blocking was performed in LI-COR Odyssey Blocking Buffer (TBS). Primary antibodies were 1:1,000 mouse anti-Flag (Sigma–Aldrich F1804) and 1:5,000 rabbit anti-gamma-Tubulin (Sigma–Aldrich T1450) in LI-COR Odyssey Blocking Buffer with 0.1% Tween. Washes were performed in TBST (TBS + 0.1% Tween). LI-COR IRDye 800CW goat anti-mouse IgG and 680RD goat anti-rabbit IgG were used at 1:15,000 in Odyssey Blocking Buffer with 0.1% Tween and 0.01% SDS. After three washes in TBST, the membranes were incubated in TBS before imaging in the LI-COR Odyssey Fc.

## Mass spectrometry and analysis

Proteins were precipitated with 23% TCA and washed with acetone. Protein pellets solubilized in 8 M urea, 100 mM Tris pH 8.5, and reduced with 5 mM Tris (2-carboxyethyl)phosphine hydrochloride (Sigma–Aldrich, St. Louis, MO, product C4706) and alkylated with 55 mM 2-chloroacetamide (Fluka Analytical, product 22790). Proteins were digested for 18 hr at 37°C in 2 M urea 100 mM Tris pH 8.5,

1 mM CaCl$_2$ with 2 µg trypsin (Promega, Madison, WI, product V5111). Single-phase analysis (in replicate) was performed using a Dionex 3000 pump and a Thermo LTQ Orbitrap Velos using an in-house built electrospray stage (*Wolters et al., 2001*). Protein and peptide identification and protein quantitation were done with Integrated Proteomics Pipeline, IP2 (Integrated Proteomics Applications, Inc, San Diego, CA; http://www.integratedproteomics.com/). Tandem mass spectra were extracted from raw files using RawConverter (*He et al., 2015*) with monoisotopic peak option and were searched against protein database release WS260 from WormBase, with FLAG-tagged SNPC-4, common contaminants and reversed sequences added, using ProLuCID (*Peng et al., 2003*; *Xu et al., 2006*). The search space included all fully tryptic and half-tryptic peptide candidates with a fixed modification of 57.02146 on C. Peptide candidates were filtered using DTASelect (*Tabb et al., 2002*).

Using custom R scripts, average enrichment between SNPC-4::3xFlag and no-tag control immunoprecipitation experiments were calculated. For each experiment, enrichment was normalized by dividing the peptide count for each protein by the total peptide count. Adjusted p-values were calculated by applying the Bonferroni method using DESeq2 (*Love et al., 2014*). Although SNPC-3.1 and SNPC-3.2 are reported to have the same amino acid sequence, we have picked up differential peptide coverage in the *fem-1(-)* mutant for these two proteins and represented them as two different data points.

## Immunofluorescence microscopy

Adult gonads were dissected into egg buffer (25 mM HEPES pH 7.4, 118 mM NaCl, 48 mM KCl, 2 mM EDTA, 0.5 mM EGTA) with 30 mM sodium azide and 0.1% Tween-20, and fixed for 10 s in 1% formaldehyde in egg buffer followed by 1 min in 100% methanol at −20℃. All washing and staining was completed in suspension. Germlines were blocked in normal goat serum or 1× Roche blocking buffer in PBST (PBS + 0.2% Tween) for 30 min at room temperature. Primary mouse anti-Flag (Sigma F1804) and rat anti-Ollas (NBP1-06713SS) antibodies were used at 1:200 in blocking agent in PBST. AlexaFluor 555 goat anti-mouse and AlexaFluor 488 goat anti-rat secondary antibodies (Thermo-Fisher) were used at 1:400 in blocking agent in PBST. Germlines were stained with 0.5 µg/mL DAPI and then mounted in Vectashield with DAPI (Vector Laboratories H-1200). Images were acquired at 63x on a Zeiss LSM700 confocal microscope. Publication images were acquired at 100× on a GE DeltaVision microscope. Image processing was performed using SoftWoRx to collect 3D image stacks, deconvolve (enhanced ratio, 20 cycles), and compile into a maximum intensity projection. Composite images were stitched and colored in Fiji using the Stitching plugin (*Preibisch et al., 2009*).

## Chromatin immunoprecipitation, library prep, and sequencing

Worms were grown in liquid culture as previously described (*Zanin et al., 2011*). 250 µM auxin was added to *snpc-1.3::3xflag; snpc-4::aid::ollas; Psun-1::TIR1* worms 4 hr before collection at 48 hr post-L1 at 20℃. After washing, the gut was cleared for 15 min by nutation in M9, followed by three washes in M9. Worms were live-crosslinked in 2.6% formaldehyde in water for 30 min at room temperature with nutation. Crosslinking was quenched with a final concentration of 125 mM glycine for 5 min with nutation. After three washes with water, worms were flash-frozen in liquid nitrogen. Frozen worm pellets were ground into powder using the Retsch MM40 ball mill homogenizer for 2 rounds of 1 min at 30 Hz. Frozen worm powder was resuspended in 1× RIPA buffer (1× PBS, 1% NP-40, 0.5% sodium deoxycholate, 0.1% SDS) for 10 min at 4℃. Crosslinked chromatin was sonicated using a Diagenode Bioruptor Pico for three 3 min cycles, 30 s on/off. We nutated 10 µg of chromatin overnight at 4℃ with 2 µg of Flag antibody (Sigma–Aldrich, F1804) and then for 1.5 hr with 50 µL mouse IgG Dynabeads (Invitrogen). Input amount was 10% of IP. Chromatin was de-crosslinked and extracted as described previously (*Weiser et al., 2017*). Individual input and IP samples of each genotype were processed for both sequencing and quantitative PCR.

Libraries were prepared and multiplexed using the Ovation Ultralow Library Systems v2 (NuGEN Technologies) according to the manufacturer's protocol. The Illumina HiSeq 4000 platform was used to generate 50 bp single-end reads for SNPC-1.3 ChIP-seq libraries. The NovaSeq 6000 platform was used to generate 50 bp paired-end reads for TRA-1 ChIP-seq libraries.

## Quantitative PCR of ChIP samples

ChIP DNA was eluted in 18 µL of $1\times$ TE pH 8.0 and 2 µL of 20 mg/mL RNase A (Invitrogen, Thermo-Fisher Scientific). For a final reaction volume of 25 µL, each reaction consisted of final $1\times$ Absolute Blue SYBR Green (ThermoFisher Scientific), 35 nM each of forward and reverse primer, and 2 µL ChIP eluate. Reactions were performed in technical duplicates in a Bio-Rad CF96 Real Time PCR thermal cycler.

## Hermaphrodite fertility assays

Gravid worms (previously maintained at 20°C) were subjected to hypochlorite treatment, and their progeny were plated onto NGM at 25°C (P0). At the L2 or L3 stage, worms were singled onto individual plates and their progeny (F1) counted.

## Mating assays

To test male-dependent rescue of *fem-1(hc17)* fertility, 10–12 hermaphrodites of each strain were grown at 20°C and embryos were isolated by allowing egg lay for 2 hr before removal. Embryos were shifted to 25°C, and upon reaching the L4 stage (24 hr), ten *him-8(e1489)* L4 males were transferred and mated with two *fem-1(hc17)* females. Brood size was quantified by counting when a majority of progeny had at least reached the young adult stage (about 3 days after transfer). To test the fertility of the hermaphrodites upon mating, 10–12 hermaphrodites of each strain were grown at 20°C and embryos were isolated after egg lay for 2 hr before removal. Embryos were shifted to 25°C and 1ten *col-19(GFP+)* L4-staged males (24 hr) were then transferred with a single hermaphrodite (36 hr), and the number of live cross progeny was counted after reaching adulthood. Brood size was quantified by counting when the majority of progeny had at least reached the young adult stage (about 3 days after transfer).

## Sperm activation assay and imaging

To perform sperm activation assays, spermatids were dissected from adult males that were shifted to 25°C during the embryo stage, and isolated prior to sexual maturity (about 48 hr post-L1). Dissection was performed directly on glass slides in sperm medium (50 mM HEPES pH 7.8, 50 mM NaCl, 25 mM KCl, 5 mM $CaCl_2$, and 1 mM $MgSO_4$) supplemented with 20 µg/mL pronase E (Millipore Sigma). For the characterization of sperm morphology, sperm were imaged 30 min after the addition of pronase E. Individual sperm were manually categorized into two types: spermatids with normal pseudopods or spermatids with irregular or no pseudopods (*Shakes and Ward, 1989*). For *Figure 6E,Z*, stacks were imaged in 10 s intervals for 30 min and a representative in-focus stack was chosen at every 3 min interval. To characterize sperm activation dynamics, sperm were individually followed across 10 s intervals for 30 min, and the different stages of sperm activation were designated into four categories based on these morphological changes: (1) undifferentiated spermatid, (2) spiky intermediate characterized by the presence of spike growth, (3) growing or motile pseudopod by the presence of a pseudopod, and (4) immobile sperm when little movement was observed either in the sperm body or pseudopod for longer than 30 s. Statistical significance was assessed using Student's t-test.

## Quantitative and statistical analysis

Unless otherwise stated, all quantitative analyses are shown as mean with standard deviation represented as error bars. For qRT-PCR, fertility and mating assays, and western blot, at least two independent experiments were performed; one representative biological replicate is shown.

## Small RNA-seq analysis

Raw small RNA-seq reads were trimmed for Illumina adapters and quality (SLIDING WINDOW: 4:25) using Trimmomatic 0.39 (*Bolger et al., 2014*). Trimmed reads were then filtered using bbmap 38.23 (http://jgi.doe.gov/data-and-tools/bb-tools) to retain reads that were 15–30 nt in length. These filtered reads were aligned to the *C. elegans* WBcel235 (*Cunningham et al., 2019*) reference genome using Bowtie 1.1.1 (*Langmead et al., 2009*) with parameters -v 0 k 5 –best –strata –tryhard. Quality control of raw and aligned reads was performed using FastQC 0.11.7 (http://www.bioinformatics.babraham.ac.uk/projects/fastqc/), SAMtools 1.9 (*Li et al., 2009*), and in-house Python and R scripts.

Mapped reads were assigned to genomic features using featureCounts from Subread 1.6.3 (*Liao et al., 2014*), taking into account overlapping and multi-mapping reads (-O -M). Raw counts were normalized within DESeq2 1.26.0 (*Love et al., 2014*), and principal component analysis (PCA) was performed using the regularized log transform of normalized counts within DESeq2 (*Figure 5—figure supplement 1C*). In addition, we distributed mapped reads by size and 5′ nucleotide identity to verify the presence of small RNA species such as 22G-RNAs and 21U-RNAs (*Figure 3—figure supplement 2B*, *Figure 5—figure supplement 1B*).

To identify differentially expressed genes, DESeq2 was applied to piRNAs on chromosome IV. In this study (method 1), we define significant and differentially expressed genes as having an absolute value of $\log_2$(fold-change) $\geq$ 0.26 and FDR of $\leq$0.05 (Benjamini–Hochberg). The $\log_2$(fold-change) threshold and significance level were selected based on benchmarking the differential expression results against the Taqman piRNA expression assays. At the chosen cutoffs, differential expression analysis captures the Taqman assays results for the three male piRNAs (21UR-1258, 21UR-3142, and 21UR-3766) and three female piRNAs (21UR-1848, 21UR-2502, and 21UR-4817). Contrasts between mutant and wild type were designed without independent filtering.

For motif discovery, nucleotide sequences were extracted from the reference genome with 60 nt upstream of each piRNA and submitted to the MEME suite 5.1.1 (*Bailey et al., 2009*). Results from MEME were used to generate the sequence logo plot with the median position of the C-nucleotide of the identified motif, number of piRNAs, and the associated E-value.

A second, independent small RNA-seq analysis workflow (described in *Figure 3—figure supplement 1*) was implemented to validate our results. Results produced from this analysis are provided in *Figure 3—figure supplement 2*. We parsed - small RNA sequences that were 16-30 nt from adapters. Reads with >3 nt falling below a quality score of Q30 were discarded. Reads were mapped to the *C. elegans* WS230 (*Stein et al., 2001*) reference genome using CASHX v. 2.3 (*Fahlgren et al., 2009*) allowing for 0 mismatches. Custom Perl, Awk, and R scripts were used to count features and to generate PCA and size distribution plots. Multi-mapping reads were assigned proportionally to each possible locus. Differential expression analysis was done using DESeq2 v. 1.18.1 (*Love et al., 2014*). A reporting threshold was set at an absolute value of $\log_2$(fold-change) $\geq$ 0.26 and a Benjamini–Hochberg-corrected p$\leq$0.20.

## ChIP-seq analysis

De-multiplexed raw ChIP-seq data in FASTQ format were trimmed for adapters and sequencing quality score > Q25 using Trim Galore! 0.5.0 (http://www.bioinformatics.babraham.ac.uk/projects/trim_galore/) and aligned to *C. elegans* reference genome WBcel235 (*Cunningham et al., 2019*) using Bowtie2 2.3.4.2 (*Langmead and Salzberg, 2012*) with default parameters. Post-alignment filtering was then performed to remove PCR duplicates using the MarkDuplicates utility within Picard 2.22.1 (http://broadinstitute.github.io/picard/). In addition, SAMtools 1.9 was applied to remove unmapped reads and reads that mapped with MAPQ 30 but were not of primary alignment or failed sequence platform quality checks (SAMtools -F 1804 -q 30) (*Li et al., 2009*).

To identify and visualize binding sites and peaks for SNPC-1.3 ChIP-seq, filtered SNPC-1.3 ChIP-seq reads were extended to 200 bp to account for the average length of ChIP fragments. We then partitioned the genome into consecutive, non-overlapping 1 kb bins and calculated read coverage, normalized by sequencing depth of each library, based on the total read count in each bin. Bins with read coverages in the IP sample that fell below the median read coverage of piRNA-depleted bins on chromosome IV in the relevant input control were excluded from further analysis. Bins containing only male, female, and non-enriched piRNAs (as defined by small RNA-seq analysis) were then extracted to generate binding profiles and heatmaps. For this, the bamCompare tool in deepTools 3.3.1 (*Ramírez et al., 2016*) was used to calculate the ratio between read coverage of each ChIP sample and input control (`–scaleFactorsMethod` None `–normalizeUsing` CPM `–operation` ratio `–binSize 50` `–ignoreForNormalization` MtDNA `–extendReads` 200). The ENCODE ce11 blacklist (*Amemiya et al., 2019*) was also supplied (https://github.com/Boyle-Lab/Blacklist/). The bamCompare output was then used in deepTools computeMatrix to calculate scores for plotting profiles and heatmaps with deepTools plotProfile and plotHeatmap.

TRA-1 ChIP-seq peaks were called by callpeak within MACS 2.1.2 (*Zhang et al., 2008*) (–p-value 0.05) with filtered TRA-1 ChIP-seq reads and relevant input controls. TRA-1 signal tracks were generated by calculating fold enrichment from read count-normalized genome-wide pileup and lambda

track outputs by callpeak (bdgcmp in MACS2). The ENCODE ce 11 blacklist (*Amemiya et al., 2019*) was supplied in this analysis (https://github.com/Boyle-Lab/Blacklist/). The bamCompare tool in deepTools 3.3.1 (*Ramírez et al., 2016*) was used to quantify read coverage of each ChIP sample and input control.

Reproducibility between SNPC-1.3 and TRA-1 ChIP-seq replicates (*Figure 4—figure supplement 1C*, *Figure 5—figure supplement 1E*) was assessed by applying deepTools bamCompare, as described above, and deepTools plotCorrelation to depict pairwise correlations between replicates and compute the Pearson correlation coefficient.

## Data and software availability

The mass spectrometry, small RNA-seq, and ChIP-seq data have been deposited in NCBI under GEO accession number: GSE152831. Processed data and scripts used for analysis are available at https://github.com/starostikm/SNPC-1.3; *Choi, 2021*; copy archived at swh:1:rev: b23f652341d999150edf5ae9c8de72e9192b2843.

## Acknowledgements

We thank Himani Galagali and Natasha Weiser for helpful comments on the manuscript. We thank members of the Kim Lab (Amelia Alessi, Mindy Clark, Gregory Fuller, Jessica Kirshner, Alex Rittenhouse, Darius Mostaghimi, Lars Benner), Tatjana Trcek, Jocelyn Haversat, Yumi Kim, Angela Andersen, Aurelia Mapps, and Jacqueline Tay for helpful suggestions. Computational resources were provided by the Maryland Advanced Research Computing Center (MARCC). Some strains were provided by the Caenorhabditis Genetics Center, which is funded by the NIH Office of Research Infrastructure Programs (P40 OD010440). This work was supported by grants from the NSF DGE-1746891 (to RJT), NIH R35 GM130272 (to SEJ); NIH R35 GM119775 (to TAM); and NIH R01 GM129301 and NIH R01 GM118875 (to JKK).

## Additional information

### Funding

| Funder | Grant reference number | Author |
| --- | --- | --- |
| National Science Foundation | DGE-1746891 | Rebecca J Tay |
| National Institute of General Medical Sciences | R35 GM130272 | Suhua Feng<br>Steven E Jacobsen |
| National Institute of General Medical Sciences | R35 GM119775 | Brooke E Montgomery<br>Taiowa A Montgomery |
| National Institute of General Medical Sciences | R01 GM12301 | Charlotte P Choi<br>Rebecca J Tay<br>Margaret R Starostik<br>Emily Xu<br>Maya A Hammonds<br>John K Kim |
| National Institute of General Medical Sciences | R01 GM118875 | Charlotte P Choi<br>Rebecca J Tay<br>Margaret R Starostik<br>Emily Xu<br>Maya A Hammonds<br>John K Kim |
| National Institute of General Medical Sciences | P41 GM103533 | James J Moresco<br>John R Yates III III |

The funders had no role in study design, data collection and interpretation, or the decision to submit the work for publication.

## Author contributions
Charlotte P Choi, Rebecca J Tay, Conceptualization, Data curation, Formal analysis, Investigation, Visualization, Methodology, Writing - original draft, Writing - review and editing; Margaret R Starostik, Data curation, Formal analysis, Visualization, Methodology, Writing - original draft; Suhua Feng, James J Moresco, Brooke E Montgomery, Emily Xu, Maya A Hammonds, John R Yates III, Steven E Jacobsen, Methodology; Michael C Schatz, Formal analysis, Supervision, Methodology; Taiowa A Montgomery, Formal analysis, Methodology; John K Kim, Conceptualization, Supervision, Funding acquisition, Methodology, Project administration, Writing - review and editing

## Author ORCIDs
Charlotte P Choi (iD) https://orcid.org/0000-0002-1857-6599
Rebecca J Tay (iD) https://orcid.org/0000-0002-2270-0505
Margaret R Starostik (iD) https://orcid.org/0000-0002-5274-2765
Taiowa A Montgomery (iD) http://orcid.org/0000-0001-7857-3253
John R Yates III (iD) https://orcid.org/0000-0001-5267-1672
John K Kim (iD) https://orcid.org/0000-0001-9838-3254

## Decision letter and Author response
Decision letter https://doi.org/10.7554/eLife.60681.sa1
Author response https://doi.org/10.7554/eLife.60681.sa2

# Additional files

### Supplementary files
• Supplementary file 1. Differential expression of piRNAs in wild-type worms during spermatogenesis and oogenesis. Related to *Figure 3*.

• Supplementary file 2. Differential expression of piRNAs in wild-type and *snpc-1.3(-)* mutants during spermatogenesis. Related to *Figure 3*.

• Supplementary file 3. Differential expression of piRNAs in wild type and *snpc-1.3 (2xtbs)* mutants during oogenesis. Related to *Figure 5*.

• Supplementary file 4. List of guide RNAs, repair templates, and oligos used in this study.

• Transparent reporting form

### Data availability
Sequencing data have been deposited in GEO under accession code GSE152831. All data generated or analyzed during this study are included in the manuscript and supporting files. Source data files have been provided for Figures 1–6.

The following dataset was generated:

| Author(s) | Year | Dataset title | Dataset URL | Database and Identifier |
|---|---|---|---|---|
| Starostik MR | 2020 | SNPC-1.3 is a sex-specific transcription factor that drives male piRNAs in *C. elegans* | https://www.ncbi.nlm.nih.gov/geo/query/acc.cgi?acc=GSE152831 | NCBI Gene Expression Omnibus, GSE152831 |

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
