## [Decision Letter]

Thank you for submitting your article "SNPC-1.3 is a sex-specific transcription factor that drives male piRNA expression in *C. elegans*" for consideration by *eLife*. Your article has been reviewed by four peer reviewers, one of whom is a member of our Board of Reviewing Editors, and the evaluation has been overseen by James Manley as the Senior Editor. The following individual involved in review of your submission has agreed to reveal their identity: Scott Kennedy (Reviewer #3).

The reviewers have discussed the reviews with one another and the Reviewing Editor has drafted this decision to help you prepare a revised submission.

Summary and Essential revisions:

All reviewers and editors think that, in principle, this manuscript is suitable for publication in *eLife*. It provides intriguing novel insights into the regulation of piRNA expression and its sexually dimorphic nature, implicating an intriguing novel player in this regulatory phenomenon. However, there was also general consensus about the need for a number of revisions. They follow essentially into two parts:

1) The only necessary experimental revision that is required to bolster claims made in the manuscript relates to the demonstration of the specificity of SNPC-1.3 expression. You will see the criticism of the current approach detailed below (particularly by reviewer #1) and the request to use available reagents (epitope tagged alleles) to attempt in situ staining. smFISH is another option. Reviewer #4, a sperm expert, was pulled into the review process late, in order to corroborate the feasibility of these studies. In further discussion among the reviewers, reviewer #4 further states: "I think the question is male germline specificity of SNPC-1.3 rather than "sperm specificity". Sperm do not express any genes but the sperm producing germline does. Presumably, males will continually express snpc-1.3 while hermaphrodites will only express around L4. You can do antibody staining on whole worms or dissected gonads. You would expect expression in the germline of males but not in the germline of adult hermaphrodites." We hope these comments are helpful.

2) The other set of revisions all deal with further data analysis and textual revisions/explanations. Reviewer #2 has the most extensive comments here and they all seem reasonable to address and require no further experimentation. All other reviewers have additional requests for editorial changes.

Reviewer #1:

The manuscript submitted by Choi et al. entitled, "SNPC-1.3 is a sex-specific transcription factor that drives male piRNA expression in *C. elegans*" describes the identification of a novel sex-specific transcriptional regulator that controls the expression of Piwi interacting RNAs (piRNAs) in a sexually dimorphic manner. The authors do an excellent job of combining biochemical, genetic, and genomic approaches to support their conclusions. While a few questions remain, overall, the findings of the manuscript represent a significant advance in the field, and therefore, warrants publication in *eLife*.

A major point of the manuscript is that a seemingly generic transcriptional complex fulfils sperm-specific functions, via the sperm-expression of a specific subunit of the complex, SNPC-1.3. However, sperm-specificity of SNPC-1.3 is not satisfactorily shown. Only a single, non-quantified, Western Blot of whole animals is shown and the reader is asked to believe that the reduction in intensity of band(s) at specific stages/in specific mutant background(s) indicates sperm-specificity. Apart from a failure of proper quantification, this is not enough. The authors have engineered two different epitope tags into the SNPC-1.3 locus and this should enable in situ antibody staining of whole animals. This should conclusively show the sperm-specificity of expression (not only sperm vs. oocyte, but also no expression in somatic tissues). Previous studies have shown the feasibility of antibody staining of sperm-expressed proteins.

Reviewer #2:

In this manuscript, Kim and colleagues have identified a Transcription Factor (TF), SNPC-1.3, specifically expressed during spermatogenesis in the hermaphroditic gonad of *C. elegans* that drives the expression of male piRNAs. SNPC-1.3 forms a specific complex with the known piRNA TF SNPC-4 during spermatogenesis. Indeed, animals lacking SNPC-1.3 specifically lose the expression of male piRNAs, even though they acquire some expression of female piRNAs. Moreover, they have identified another TF, TRA-1, which restricts the expression of SNPC-1.3 exclusively during spermatogenesis. TRA-1 is expressed during oogenesis and binds the snpc-1.3 promoter to silence the expression of SNPC-1.3 during oogenesis. Thus, mutation of the binding sites of TRA-1 leads to ectopic expression of male piRNAs during spermatogenesis. Finally, snpc-1.3 mutation leads to spermatogenesis defects and loss of fertility similarly to piRNA mutants, suggesting a role for male piRNAs in spermatogenesis. Overall the experiments are well done and executed with appropriate controls. Even though the function of these male and female piRNAs are still uncharacterized, the study is certainly of general interests given the fact that is currently unknown how the transcription of male and/or female piRNAs are specifically achieved. The manuscript is well written, and the data are well presented, although some additional controls, experiments, and edits might help to solidify their findings. My only major concerns regard the specificity of binding and action of SNPC-1.3 for male-specific piRNAs. Given that mutation of sncp-1.3 affects both male and female piRNAs during spermatogenesis and there are not clear evidences of binding by ChIP of SNCP-1.3 on male piRNA promoters vs. the other piRNAs, it remains unclear from the present data how SNPC-1.3 drives the exclusive expression of male piRNAs, and how SNPC-1.3 cooperates with SNPC-4 to specifically bind the promoters of male piRNAs and not female piRNAs. The localization of SNPC-1.3, SNPC-4, and TRA-1 in hermaphrodite and male germlines might help to elucidate their temporal and sex-specific functions together with the specific binding of male piRNA consensus sequence. Below I have listed the specific comments that the authors should address in order that the paper may be considered for publication on *eLife*.

1) As I mention above, it remains unclear how SNPC-3.1 specifically binds the promoter of male piRNAs. The authors should evaluate whether the ChIP-seq signal is enriched around the consensus motif of male piRNAs and not the female (or neutral NE) piRNAs. Also, it is not clear how SNPC-1.3 provides the specificity to SNPC-4 in transcribing only male piRNAs. For instance, they have mentioned in the Discussion that SNPC-4 can still bind piRNA loci in absence of SNPC-1.3 (the authors should also provide such data, see other points). Moreover, in the Discussion they mentioned that they "cannot definitively conclude that SNPC-1.3 binds to the male-specific core motif, given the limitations of conventional ChIP-seq in resolving the SNPC-1.3 footprint. Identifying the factors that specifically bind the 8 nt core motif and other potential *cis*-regulatory elements important for sex-biased piRNA expression will require further investigation". I think the authors can at least assess whether SNPC-1.3 binds the male consensus motif and not the female one in vitro by EMSA.

2) In absence of any specific binding of SNPC-1.3 to male/spermatogenic piRNAs the only experiment that suggests such specific function is the piRNA quantification in *snpc-1.3(-)* mutants. However, as they mentioned in their manuscript: "oogenesis-enriched piRNAs were upregulated at least 2-fold in *snpc-1.3(-)* mutants undergoing spermatogenesis and in *him-8(-)*; *snpc-1.3(-)* males. These findings suggest that, in addition to activating male piRNAs, SNPC-1.3 suppresses the expression of female piRNAs in the male germline, possibly by preferentially recruiting core factors such as SNPC-4 to male piRNA loci". This model should be demonstrated by SNPC-4 ChIP in sncp-1.3(-) mutants. However, they mention in the Discussion that sncp-1.3(-) mutants do not affect SNPC-4 binding. So, this is unlikely the case. An alternative explanation of these results might be that mutation in sncp-1.3 caused some developmental defects in the temporal switching between spermatogenesis and oogenesis. Also, the ChIP signal of SNPC-1.3 do not appear to occur specifically on male piRNA, but also on neutral non-enriched (NE) piRNAs and to some degree to female piRNAs (Figure 3E). Even in the context of tra-1 mutation there is not an exclusive upregulation of male piRNA, but also downregulation of female piRNA. Therefore, the specificity of SNPC-3.1 for male piRNA transcription is not clear.

3) Throughout the whole manuscript the authors wrote that SNPC-1.3 drives the expression of male piRNAs. However, the experiments have been conducted in hermaphrodite germlines (or mutants) and not in male germlines. Maybe it is more correct to talk about spermatogenic expressed/enriched and oogenic expressed/enriched piRNAs instead of male/female piRNAs. In this regard, the authors mention that SNPC-1.3 is expressed during spermatogenesis and is specific for male. But, this has been only deducted by RNA-seq at different time point of hermaphroditic germline development. Ideally the authors should create a fluorescent tag version of SNPC-1.3 to evaluate its specific temporal expression in hermaphrodite as well as in male. If this is not possible the authors should perform immunostaining using FLAG antibody. This can be combined with SNPC-4 and TRA-1 staining. Also, it would be interesting to evaluate the expression of PIWI by staining and/or WB during spermatogenesis and oogenesis in *snpc-1.3(-)* mutants.

4) In the legend of Figure 3 it is written "SNPC-1.3::3xFlag binding normalized to input (mean ± SD of two technical replicates) on chromosome IV by ChIP-qPCR in a no-tag control, the strain expressing SNPC-4::3xFlag (wildtype), and in the strain expressing SNPC-4::3xFlag::AID, which undergoes TIR-1-mediated degradation by addition of auxin (*snpc-4::aid*)". This is very confusing and if it is true it means that the SNPC-1.3 ChIP can detect also the binding of SNPC-4 (they are both tagged with FLAG), hence the specific binding of SNPC-3.1 vs. SNPC-4 cannot be determined. Maybe there is a mistake in the legend and the authors should clarify this (it was difficult for me to find this information in the strain list). Also, to avoid such confusion the authors should not call wild-type an edited strain and all the information about the strain should be clearer in the figures.

5) In the Discussion, the authors wrote "Due to the highly clustered nature of *C. elegans* piRNAs, we have been unable to discriminate detectable differences in SNPC-4 binding between male and female piRNAs in *snpc-1.3(-)* mutants, as assayed by traditional ChIP-seq methods." I was unable to find these data in the manuscript. Are they present in another publication? Or those are data not shown? I think it would be important to add these data and comment on why only SNPC-4 is required for SNPC-1.3 binding and not vice versa. Also, in case these data are included, the authors should comment on the point I raised above.

6) In Figure 2A the levels of male piRNAs is not completely abolished (like in piwi mutant). The authors should comment on this and whether they think there are additional TF factors that drive the expression of male piRNAs. This is also evident in the small RNA-seq experiment in Figure 2D. Here, not all the male piRNA are depleted at similar levels. Maybe the authors can look at the consensus motif of the most affected vs. less affected piRNA groups to see if they find any differences. Also, it would be interesting to analyze the expression levels of the most affected ones compared to the less affected ones to verify that the severity of the loss is not caused by their relative abundance.

7) In Figure 5A the authors show that snpc-3.1 mutant has reduced brood size at 25°C similarly to *prg-1* mutant. However, *prg-1* mutant is not shown for comparison. Also, given that *prg-1* mutant shows progressive sterility or mortal germline at 20°C and 25°C, it would be important to know at which generation they have performed their experiment and whether snpc-3.1 mutant is also mortal. Finally, they need to evaluate the fertility at 20°C as well as at 25°C to have a good comparison with *prg-1* mutant.

Reviewer #3:

piRNAs are major regulators of gene expression and transposable elements in animal germlines. Many aspects of piRNA biology are sexually dimorphic and the mechanistic underpinnings of this dimorphism are largely unknown. In this work, Choi et al. provide compelling evidence that they have defined the mechanism by which some piRNAs are expressed specifically in the male germline of the nematode *C. elegans*. The subject matter is of broad interest. The paper is very well written. The data are of very high quality. The conclusions are well supported by the data. I recommend publishing the work in *eLife*.

---

## [Author Response]

All reviewers and editors think that, in principle, this manuscript is suitable for publication in eLife. It provides intriguing novel insights into the regulation of piRNA expression and its sexually dimorphic nature, implicating an intriguing novel player in this regulatory phenomenon. However, there was also general consensus about the need for a number of revisions. They follow essentially into two parts:1) The only necessary experimental revision that is required to bolster claims made in the manuscript relates to the demonstration of the specificity of SNPC-1.3 expression. You will see the criticism of the current approach detailed below (particularly by reviewer #1) and the request to use available reagents (epitope tagged alleles) to attempt in situ staining. smFISH is another option. Reviewer #4, a sperm expert, was pulled into the review process late, in order to corroborate the feasibility of these studies. In further discussion among the reviewers, reviewer #4 further states: "I think the question is male germline specificity of SNPC-1.3 rather than "sperm specificity". Sperm do not express any genes but the sperm producing germline does. Presumably, males will continually express snpc-1.3 while hermaphrodites will only express around L4. You can do antibody staining on whole worms or dissected gonads. You would expect expression in the germline of males but not in the germline of adult hermaphrodites." We hope these comments are helpful.

Our major revisions aim to demonstrate that SNPC-1.3 is predominantly expressed in the male germline. We have included the following data in a new Figure 2:

new Figure 2:

Figure 2A. RT-qPCR and western blotting analyses in germline-less worms using RNAi knockdown and mutants show that *snpc-1.3* mRNA and protein are mainly expressed in the germline.

Figure 2B: *snpc-1.3* mRNA and protein expression in *him-8(-)* males and *fem-1(-)* females shows SNPC-1.3 enrichment in the male germline.

Figure 2C: Immunofluorescence of SNPC-1.3 and SNPC-4 demonstrate that these two factors colocalize into nuclear foci in the male, but not hermaphrodite, germline.

2) The other set of revisions all deal with further data analysis and textual revisions/explanations. Reviewer #2 has the most extensive comments here and they all seem reasonable to address and require no further experimentation. All other reviewers have additional requests for editorial changes.

We have extensively addressed these other comments raised by the reviewers. Please find our detailed, point-by-point responses to each of the reviewers’ comments below. Collectively, we believe our revisions, based on the reviewers’ comments, strengthen our study and feel that it is now acceptable for publication in *eLife*.

Reviewer #1:The manuscript submitted by Choi et al. entitled, "SNPC-1.3 is a sex-specific transcription factor that drives male piRNA expression in *C. elegans*" describes the identification of a novel sex-specific transcriptional regulator that controls the expression of Piwi interacting RNAs (piRNAs) in a sexually dimorphic manner. The authors do an excellent job of combining biochemical, genetic, and genomic approaches to support their conclusions. While a few questions remain, overall, the findings of the manuscript represent a significant advance in the field, and therefore, warrants publication in eLife.A major point of the manuscript is that a seemingly generic transcriptional complex fulfils sperm-specific functions, via the sperm-expression of a specific subunit of the complex, SNPC-1.3. However, sperm-specificity of SNPC-1.3 is not satisfactorily shown. Only a single, non-quantified, Western Blot of whole animals is shown and the reader is asked to believe that the reduction in intensity of band(s) at specific stages/in specific mutant background(s) indicates sperm-specificity. Apart from a failure of proper quantification, this is not enough.

We appreciate the reviewer’s concerns and hope we have addressed them by clarifying the text for the existing figures and including a new Figure 2. Specifically, Figure 2 now includes SNPC-1.3 expression data at both the mRNA and protein level, with quantification, in males and females.

The authors have engineered two different epitope tags into the SNPC-1.3 locus and this should enable in situ antibody staining of whole animals. This should conclusively show the sperm-specificity of expression (not only sperm vs. oocyte, but also no expression in somatic tissues). Previous studies have shown the feasibility of antibody staining of sperm-expressed proteins.

We agree and now provide this experiment in Figure 2C, which now includes antibody staining of dissected adult germlines in males and hermaphrodites. While whole-mount staining of SNPC-1.3 would be informative, we have addressed SNPC-1.3 specificity in the germline by examining *snpc-1.3* mRNA and protein levels via *glp-1* RNAi and a *glp-4* mutant in Figure 2A.

Reviewer #2:In this manuscript, Kim and colleagues have identified a Transcription Factor (TF), SNPC-1.3, specifically expressed during spermatogenesis in the hermaphroditic gonad of *C. elegans* that drives the expression of male piRNAs. SNPC-1.3 forms a specific complex with the known piRNA TF SNPC-4 during spermatogenesis. Indeed, animals lacking SNPC-1.3 specifically lose the expression of male piRNAs, even though they acquire some expression of female piRNAs. Moreover, they have identified another TF, TRA-1, which restricts the expression of SNPC-1.3 exclusively during spermatogenesis. TRA-1 is expressed during oogenesis and binds the snpc-1.3 promoter to silence the expression of SNPC-1.3 during oogenesis. Thus, mutation of the binding sites of TRA-1 leads to ectopic expression of male piRNAs during spermatogenesis. Finally, snpc-1.3 mutation leads to spermatogenesis defects and loss of fertility similarly to piRNA mutants, suggesting a role for male piRNAs in spermatogenesis. Overall the experiments are well done and executed with appropriate controls. Even though the function of these male and female piRNAs are still uncharacterized, the study is certainly of general interests given the fact that is currently unknown how the transcription of male and/or female piRNAs are specifically achieved. The manuscript is well written, and the data are well presented, although some additional controls, experiments, and edits might help to solidify their findings. My only major concerns regard the specificity of binding and action of SNPC-1.3 for male-specific piRNAs. Given that mutation of sncp-1.3 affects both male and female piRNAs during spermatogenesis and there are not clear evidences of binding by ChIP of SNCP-1.3 on male piRNA promoters vs. the other piRNAs, it remains unclear from the present data how SNPC-1.3 drives the exclusive expression of male piRNAs, and how SNPC-1.3 cooperates with SNPC-4 to specifically bind the promoters of male piRNAs and not female piRNAs. The localization of SNPC-1.3, SNPC-4, and TRA-1 in hermaphrodite and male germlines might help to elucidate their temporal and sex-specific functions together with the specific binding of male piRNA consensus sequence. Below I have listed the specific comments that the authors should address in order that the paper may be considered for publication on eLife.1) As I mention above, it remains unclear how SNPC-3.1 specifically binds the promoter of male piRNAs. The authors should evaluate whether the ChIP-seq signal is enriched around the consensus motif of male piRNAs and not the female (or neutral NE) piRNAs.

We appreciate the reviewer’s comment. Given that we did not have the necessary nucleotide resolution in our ChIP-seq data to definitely discriminate whether SNPC-1.3 binds to the promoter of male piRNAs, we acknowledged this caveat and stated the following in our first submission:

“While we observed SNPC-1.3 binding to be enriched upstream of male piRNA loci (Figure 4), we cannot definitively conclude that SNPC-1.3 binds to the male-specific core motif, given the limitations of conventional ChIP-seq in resolving the SNPC-1.3 footprint. Identifying the factors that specifically bind the 8 nt core motif and other potential *cis*-regulatory elements important for sex-specific piRNA expression will require further investigation.”

Also, it is not clear how SNPC-1.3 provides the specificity to SNPC-4 in transcribing only male piRNAs. For instance, they have mentioned in the Discussion that SNPC-4 can still bind piRNA loci in absence of SNPC-1.3 (the authors should also provide such data, see other points).

We thank the reviewer for this point and agree that having SNPC-4 ChIP would enhance our understanding. Unfortunately, while we have tried very hard, we have not yet been able to successfully complete this experiment: ChIP signals for SNPC-4 are notoriously weak and therefore we did not include these experiments. However, we do not think the lack of SNPC-4 ChIP data in *snpc-1.3(-)* significantly detracts from answering the major hypotheses of this paper. In the Discussion, we have made sure to emphasize that we do not yet have conclusive evidence that SNPC-4 binding to piRNA promoters requires SNPC-1.3.

Moreover, in the Discussion they mentioned that they "cannot definitively conclude that SNPC-1.3 binds to the male-specific core motif, given the limitations of conventional ChIP-seq in resolving the SNPC-1.3 footprint. Identifying the factors that specifically bind the 8 nt core motif and other potential cis-regulatory elements important for sex-biased piRNA expression will require further investigation". I think the authors can at least assess whether SNPC-1.3 binds the male consensus motif and not the female one in vitro by EMSA.

We thank the reviewers for their suggestion. While an EMSA, in theory, is a good way to test specificity of the male motif, we do not find it feasible for this paper. Specifically, in vitro work done using fly cells shows that SNPC-1.3 binding can only occur within the context of the entire SNPC1/3/4 complex. Therefore, we decided against performing these types of experiments because we expect that an EMSA may not be feasible unless we also purify multiple SNAP complex proteins, as well as other piRNA factors (such as the TOFUs and PRDE-1). Additionally, the presence of the core motif may not be sufficient for SNPC-1.3 binding.

2) In absence of any specific binding of SNPC-1.3 to male/spermatogenic piRNAs the only experiment that suggests such specific function is the piRNA quantification in snpc-1.3(-) mutants. However, as they mentioned in their manuscript: "oogenesis-enriched piRNAs were upregulated at least 2-fold in snpc-1.3(-) mutants undergoing spermatogenesis and in him-8(-); snpc-1.3(-) males. These findings suggest that, in addition to activating male piRNAs, SNPC-1.3 suppresses the expression of female piRNAs in the male germline, possibly by preferentially recruiting core factors such as SNPC-4 to male piRNA loci". This model should be demonstrated by SNPC-4 ChIP in sncp-1.3(-) mutants. However, they mention in the Discussion that sncp-1.3(-) mutants do not affect SNPC-4 binding. So, this is unlikely the case.

Please see our previous comments on the feasibility of SNPC-4 ChIP. We have edited the Discussion to stress that the perturbation of SNPC-4 binding in *snpc-1.3(-)* is our working hypothesis and requires further investigation in future studies.

An alternative explanation of these results might be that mutation in sncp-1.3 caused some developmental defects in the temporal switching between spermatogenesis and oogenesis. Also, the ChIP signal of SNPC-1.3 do not appear to occur specifically on male piRNA, but also on neutral non-enriched (NE) piRNAs and to some degree to female piRNAs (Figure 3E). Even in the context of tra-1 mutation there is not an exclusive upregulation of male piRNA, but also downregulation of female piRNA. Therefore, the specificity of SNPC-3.1 for male piRNA transcription is not clear.

We thank the reviewers for these observations and agree that the loss of male piRNAs could be a consequence of increased female piRNA expression, but technical limitations make this difficult to test. While it is plausible that *snpc-1.3* is involved in the sexual identity switch of the germline, we think this is highly unlikely as we do not observe any morphological defects that would suggest transformation of sexual identity from male to female in *snpc-1.3* mutants.

In reference to the lack of specificity of SNPC-1.3 binding to male, female and NE piRNAs, we think this could be due to the technical limitations of our ChIP datasets. We have plotted the distance of the nearest neighboring piRNA for each piRNA in the two chromosome IV clusters (Author response image 1). As the distances between piRNAs fall below the average footprint obtained by conventional ChIP-seq methods, we were unable to identify distinct binding sites for individual piRNAs.

**Author response image 1. respfig1:** Distance of the nearest neighboring piRNA for each piRNA in the two chromosome IV clusters.

The SNAP complex is well defined as a transcription activator. We therefore believe that female piRNA repression is a result of sequestration of SNPC-4 by the male piRNA promoters rather than direct repression.We feel the novelty of our paper is based on the observation that SNPC-1.3-dependent regulation is required for the balance between male and female piRNAs. If the resolution of our ChIP was higher (e.g. to the level of Cut-N-Run or ChIP-exo), we hypothesize that we would be able to address whether or not SNPC-1.3 binds directly to male piRNA promoters.

3) Throughout the whole manuscript the authors wrote that SNPC-1.3 drives the expression of male piRNAs. However, the experiments have been conducted in hermaphrodite germlines (or mutants) and not in male germlines. Maybe it is more correct to talk about spermatogenic expressed/enriched and oogenic expressed/enriched piRNAs instead of male/female piRNAs. In this regard, the authors mention that SNPC-1.3 is expressed during spermatogenesis and is specific for male. But, this has been only deducted by RNA-seq at different time point of hermaphroditic germline development.

We thank the reviewer for these points and acknowledge that these terms can often be confusing. However, we would like to keep the existing designation of “male/female” piRNAs as we feel that the use “spermatogenic/oogenic” piRNAs would create additional confusion when describing piRNA populations that are inappropriately expressed during spermatogenesis and oogenesis in certain *snpc-1.3* mutants. We also feel that this designation is appropriate as we used this terminology in our previous study (Billi et al., 2013), which validated that “male” and “female” piRNAs discovered by sequencing in male and female germlines are indeed expressed in hermaphrodites during spermatogenesis and oogenesis, respectively, by Taqman qPCR.

Ideally the authors should create a fluorescent tag version of SNPC-1.3 to evaluate its specific temporal expression in hermaphrodite as well as in male. If this is not possible the authors should perform immunostaining using FLAG antibody. This can be combined with SNPC-4 and TRA-1 staining.

Done. We have examined SNPC-1.3 and SNPC-4 expression by immunostaining in the new Figure 2C, which shows that SNPC-1.3 co-localizes with SNPC-4 and is detectable only in male germline nuclei.

Also, it would be interesting to evaluate the expression of PIWI by staining and/or WB during spermatogenesis and oogenesis in snpc-1.3(-) mutants.

Previous studies have shown that PRG-1 localizes to perinuclear granules and its function in the piRNA RISC complex is likely downstream of piRNA transcription. Nevertheless, while evaluating PRG-1 expression in the context of *snpc-1.3(-)* mutants could be interesting, we feel that this analysis is outside the scope of this study.

4) In the legend of Figure 3 it is written "SNPC-1.3::3xFlag binding normalized to input (mean ± SD of two technical replicates) on chromosome IV by ChIP-qPCR in a no-tag control, the strain expressing SNPC-4::3xFlag (wildtype), and in the strain expressing SNPC-4::3xFlag::AID, which undergoes TIR-1-mediated degradation by addition of auxin (snpc-4::aid)". This is very confusing and if it is true it means that the SNPC-1.3 ChIP can detect also the binding of SNPC-4 (they are both tagged with FLAG), hence the specific binding of SNPC-3.1 vs. SNPC-4 cannot be determined.

We apologize for this error in the text. The experiments were performed in *snpc-1.3::3xflag* worms in the N2 and *snpc-4::aid::ollas* backgrounds. We have corrected this in the legend.

Maybe there is a mistake in the legend and the authors should clarify this (it was difficult for me to find this information in the strain list). Also, to avoid such confusion the authors should not call wild-type an edited strain and all the information about the strain should be clearer in the figures.

We thank the reviewers for this point. The legends, figures, and text have been changed to clarify the genetic background of each strain.

5) In the Discussion, the authors wrote "Due to the highly clustered nature of *C. elegans* piRNAs, we have been unable to discriminate detectable differences in SNPC-4 binding between male and female piRNAs in snpc-1.3(-) mutants, as assayed by traditional ChIP-seq methods." I was unable to find these data in the manuscript. Are they present in another publication? Or those are data not shown? I think it would be important to add these data and comment on why only SNPC-4 is required for SNPC-1.3 binding and not vice versa. Also, in case these data are included, the authors should comment on the point I raised above.

We apologize for this confusion. We have not been able to generate our own SNPC-4 ChIP +/- *snpc-1.3* in this manuscript. We have edited the text to stress that we anticipate resolving changes in SNPC-4 binding in *snpc-1.3(-)* will be difficult.

6) In Figure 2A the levels of male piRNAs is not completely abolished (like in piwi mutant). The authors should comment on this and whether they think there are additional TF factors that drive the expression of male piRNAs.

We agree with the reviewers that it is possible that other male transcription factors exist. However, we think the slightly elevated expression of male piRNA expression in *snpc-1.3* compared to *prg-1* mutants (Figure 3A) is likely a result of leaky transcription at piRNA loci. Given that PRG-1 is the major Argonaute that loads all piRNAs downstream of biogenesis, we do not find it surprising that *prg-1* mutants exhibit more severe loss of piRNAs compared to *snpc-1.3* mutants.

This is also evident in the small RNA-seq experiment in Figure 2D. Here, not all the male piRNA are depleted at similar levels. Maybe the authors can look at the consensus motif of the most affected vs. less affected piRNA groups to see if they find any differences.

Our motif analysis has demonstrated that the overwhelming majority of SNPC-1.3-dependent male piRNAs have the 5’C motif (3,429/3,452).

Also, it would be interesting to analyze the expression levels of the most affected ones compared to the less affected ones to verify that the severity of the loss is not caused by their relative abundance.

We thank the reviewers for this suggestion. To test if the degree of male piRNA expression changes seen in *snpc-1.3(-)* mutants are correlated to their relative abundance during wildtype spermatogenesis, we plotted the absolute value of the log_2_(fold-change) of each male piRNA depleted in *snpc-1.3(-)* mutants against its mean expression level during spermatogenesis in wildtype (Author response image 2). We conclude from this additional analysis that, overall, piRNAs with greater relative abundance in wildtype do show greater depletion in *snpc-1.3(-)* mutants during spermatogenesis.

**Author response image 2. respfig2:** Absolute value of the log_2_(fold-change) of each male piRNA depleted in *snpc-1.3(-)* mutants against its mean expression level during spermatogenesis in wildtype.

7) In Figure 5A the authors show that snpc-3.1 mutant has reduced brood size at 25°C similarly to prg-1 mutant. However, prg-1 mutant is not shown for comparison. Also, given that prg-1 mutant shows progressive sterility or mortal germline at 20°C and 25°C, it would be important to know at which generation they have performed their experiment and whether snpc-3.1 mutant is also mortal. Finally, they need to evaluate the fertility at 20°C as well as at 25°C to have a good comparison with prg-1 mutant.

We thank the reviewers for bringing up this point. We have made sure to include in the legends and Materials and methods that all brood count assays were shifted and assayed at the P0 generation at 25°C in Figure 6. We were careful not to make any direct comparisons between *snpc-1.3(-)* and *prg-1(-)* fertility, but we did want to show that *snpc-1.3(-)* mutants exhibit significant fertility defects. We also see a significant but attenuated fertility defect in *snpc-1.3(-)* at 20°C (see Author response image 3). Although we agree with the reviewers that investigating the transgenerational effects of sex-specific piRNAs could be very interesting, we think this area is beyond the scope of this paper. However, we hope to characterize sex-specific transgenerational effects in future studies.

**Author response image 3. respfig3:** Fertility defect in *snpc-1.3(-)* at 20°C.